# MafB is a critical regulator of complement component C1q

Mai Thi Nhu Tran[1], Michito Hamada [1,2], Hyojung Jeon[1], Risako Shiraishi[1], Keigo Asano[1], Motochika Hattori[1], Megumi Nakamura[1], Yuki Imamura[1], Yuki Tsunakawa[1,3], Risa Fujii[1], Toshiaki Usui[1], Kaushalya Kulathunga[1,3], Christina-Sylvia Andrea[4], Ryusuke Koshida[1], Risa Kamei[1], Yurina Matsunaga[1], Makoto Kobayashi [4], Hisashi Oishi[5], Takashi Kudo[1,2] & Satoru Takahashi[1,2,6,7,8]

The transcription factor MafB is expressed by monocytes and macrophages. Efferocytosis (apoptotic cell uptake) by macrophages is important for inhibiting the development of autoimmune diseases, and is greatly reduced in *Mafb*-deficient macrophages. Here, we show the expression of the first protein in the classical complement pathway C1q is important for mediating efferocytosis and is reduced in *Mafb*-deficient macrophages. The efferocytosis defect in *Mafb*-deficient macrophages can be rescued by adding serum from wild-type mice, but not by adding serum from C1q-deficient mice. By hemolysis assay we also show that activation of the classical complement pathway is decreased in *Mafb*-deficient mice. In addition, MafB overexpression induces C1q-dependent gene expression and signals that induce C1q genes are less effective in the absence of MafB. We also show that *Mafb*-deficiency can increase glomerular autoimmunity, including anti-nuclear antibody deposition. These results show that MafB is an important regulator of C1q.

[1] Department of Anatomy and Embryology, Faculty of Medicine, University of Tsukuba, 1-1-1, Tennodai, Tsukuba, Ibaraki 305-8575, Japan. [2] Laboratory Animal Resource Center, University of Tsukuba, 1-1-1, Tennodai, Tsukuba, Ibaraki 305-8575, Japan. [3] Ph.D. Program in Human Biology, School of Integrative and Global Majors, University of Tsukuba, 1-1-1, Tennodai, Tsukuba, Ibaraki 305-8577, Japan. [4] Department of Molecular and Developmental Biology, Faculty of Medicine, University of Tsukuba, 1-1-1, Tennodai, Tsukuba, Ibaraki 305-8575, Japan. [5] Department of Comparative and Experimental Medicine, Nagoya City University Graduate 24 School of Medical Sciences, 1 Kawasumi, Mizuho-cho, Mizuho-ku, Nagoya 467-8601, Japan. [6] International Institute for Integrative Sleep Medicine (WPI-IIIS), University of Tsukuba, 1-1-1, Tennodai, Tsukuba, Ibaraki 305-8575, Japan. [7] Transborder Medical Research Center, University of Tsukuba, 1-1-1, Tennodai, Tsukuba, Ibaraki 305-8577, Japan. [8] Life Science Center, Tsukuba Advanced Research Alliance, University of Tsukuba, 1-1-1, Tennodai, Tsukuba, Ibaraki 305-8577, Japan. Mai Thi Nhu Tran, Michito Hamada, Hyojung Jeon, and Risako Shiraishi contributed equally to this work. Correspondence and requests for materials should be addressed to M.H. (email: hamamichi@md.tsukuba.ac.jp) or to S.T. (email: satoruta@md.tsukuba.ac.jp)

MafB is a member of the large Maf transcription factor family, which includes basic-region leucine zipper type transcriptional factors that bind to the Maf recognition element (MARE) by dimerization[1]. MafB is expressed by, and is important for the differentiation of, a variety of cell types, such as pancreatic α and β cells, podocytes in the renal glomerulus, and rhombomere r5 in the embryonic hindbrain, thymus, parathyroid gland, and hair cuticles[2–6]. In haematopoietic cells, MafB is also expressed selectively by monocytes and macrophages. Although MafB is dispensable for macrophage differentiation[3, 7, 8], MafB inhibits foam cell apoptosis in atherosclerotic lesions by reducing the expression of apoptosis inhibitor of macrophage (AIM)[9], indicating that MafB is important for the physiological properties of macrophages and that it is possible MafB regulates macrophage function in other diseases.

C1q binds to antigen-bound antibody molecules, leading to activation of the classical complement pathway. C1q consists of three subunits (C1qA, C1qB, and C1qC), and binds to a variety of ligands and regulates biological and cellular responses through two domains, the gC1q (globular head) region and the collagen-like region[10]. C1q binds phosphatidylserine on early apoptotic cells via gC1q and then prompts efferocytosis by macrophages. Apoptotic cells are a source of self-antigen and lead to the production of autoantibodies; therefore, *C1qa*-deficient mice spontaneously develop systemic lupus erythematosus (SLE)-like disease due to the accumulation of apoptotic cells[11]. Patients with C1q deficiency also have >90% probability of developing SLE or SLE-like autoimmune diseases. Homozygous C1q deficiency is infrequent, but this trait is the strongest genetic risk factor for SLE[11–13]. C1q expression is restricted primarily to macrophages

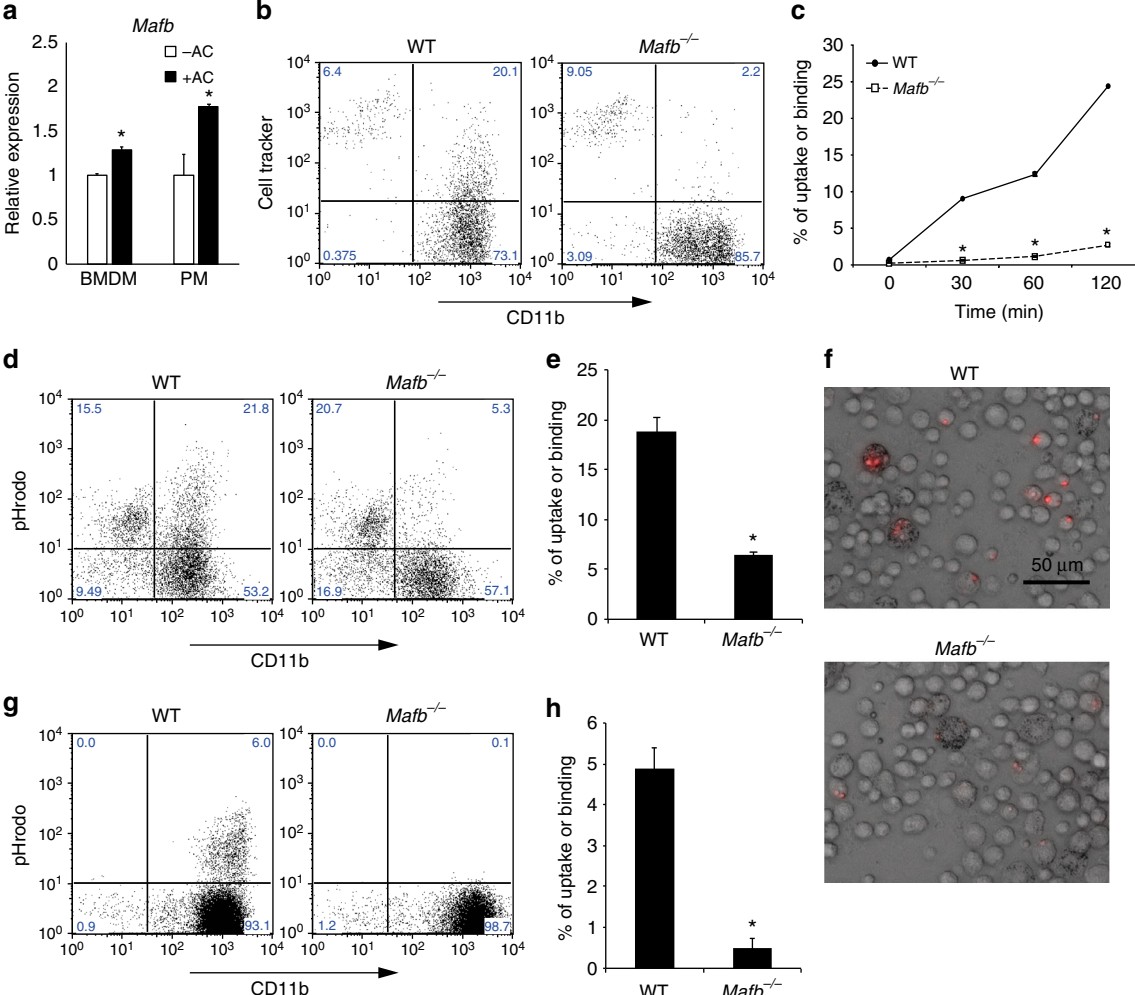

**Fig. 1** *Mafb*-deficient macrophages are unable to engulf apoptotic cells. **a** Apoptotic cell-induced expression of *Mafb* in both BMDMs and PMs. *Mafb* mRNA was quantified by qRT-PCR; $n = 4$ for each group. The data were normalized to *Hprt* mRNA levels and are presented as the mean ± s.e.m., *$p < 0.05$ compared with apoptotic cells (−AC) (Student's *t*-test). The data are from one experiment representative of two independent experiments. **b**, **c** Jurkat T cells were induced to apoptosis and labeled with CellTracker. Apoptotic cells were incubated with WT and *Mafb*−/− macrophages. CD11b and CellTracker double-positive populations represent macrophages that bind and/or engulf apoptotic cells. **c** The percentage of binding or uptake of apoptotic cells was increased in a time-dependent manner in WT but not *Mafb*−/− (WT, $n = 5$; *Mafb*−/−, $n = 6$). **d**, **e** Apoptotic thymocytes were stained with pHrodo and incubated with fetal liver-derived macrophages for 120 min. **e** The percentage of cells that were taken up was significantly reduced in *Mafb*−/− macrophages ($n = 5$ for each group). **g**, **h**, **f** Peritoneal exudate cells (PECs) were analyzed by FACS after apoptotic thymocytes treated with pHrodo were injected into the mice, which were injected with thioglycolate 3 days before. **h** The percentage of apoptotic cell uptake was significantly reduced in *Mafb*−/− macrophages (WT, $n = 4$; *Mafb*−/−, $n = 5$). **f** Microscopic analysis showed that pHrodo fluorescence was observed in the PECs of WT, but not *Mafb*−/−. **c**, **e**, **h** Quantification data are presented as the mean ± s.e.m. *$p < 0.05$ compared with WT, (Student's *t*-test). The data are from one experiment that was representative of at least two independent experiments

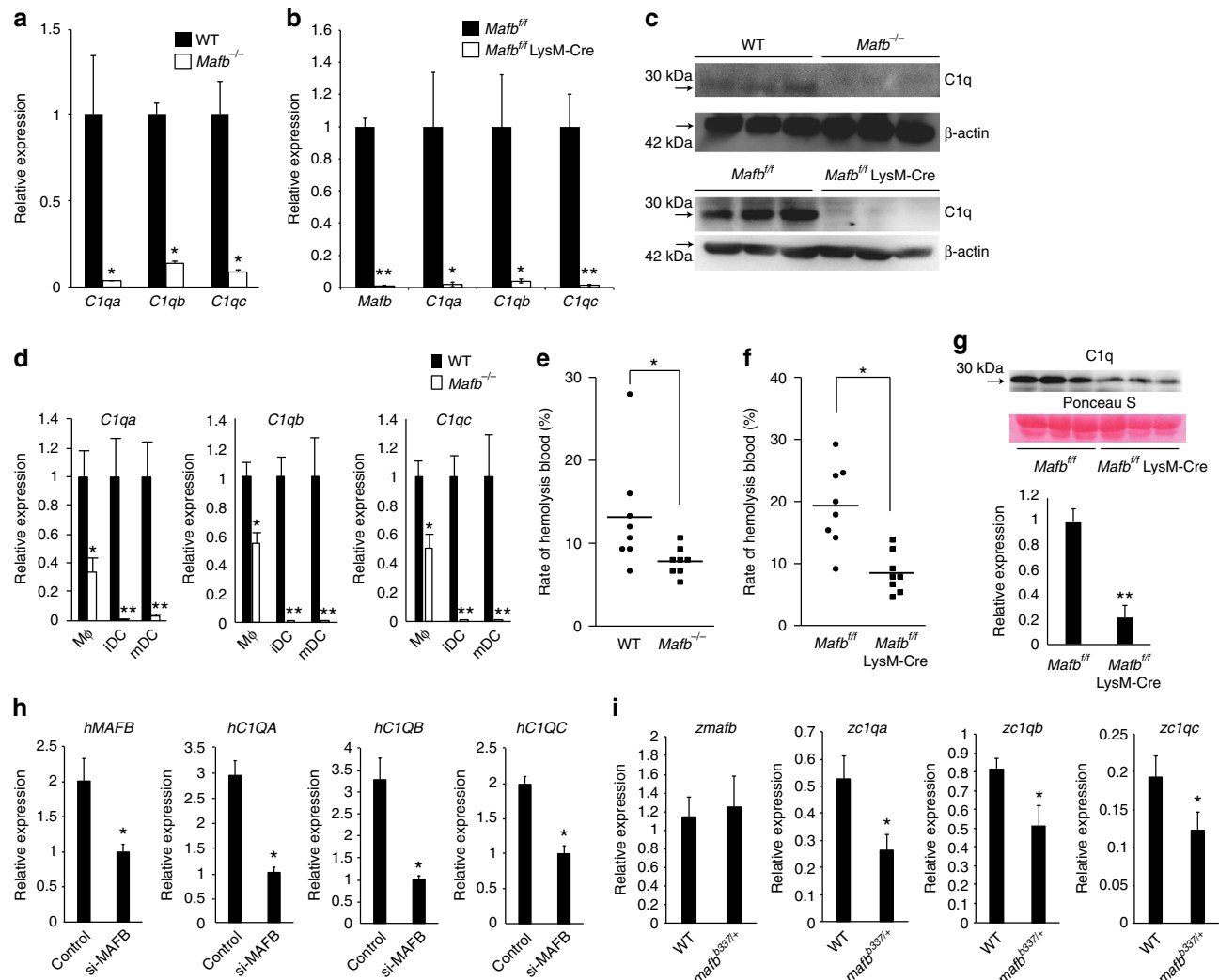

**Fig. 2** C1q gene expression is decreased in *Mafb*[−/−] macrophages. **a** Messenger RNAs of C1q genes in PMs were examined by qRT-PCR (*n* = 3 for each group). **b** PMs of *Mafb*[f/f]::LysM-Cre and *Mafb*[f/f] were analyzed for the expression of the *Mafb* and C1q genes (*n* = 4 for each group). **c** A western blot analysis was performed to measure C1q protein levels in the PMs of WT and *Mafb*[−/−]. The protein extracts were normalized to β-actin expression. The data are from one experiment that was representative of at least two independent experiments. **d** The C1q genes were analyzed by qRT-PCR of macrophages and immature DCs from WT or *Mafb*[−/−] (*n* = 4 for each group). The data are from one experiment that was representative of at least two independent experiments. **a**, **b**, **d** The data were normalized to *Hprt* mRNA and are presented as the mean ± s.e.m. *\*p < 0.05* compared with WT, *\*\*p < 0.01* (Student's *t*-test). **e**, **f** The hemolysis rates were analyzed using serum from fetal liver cell-transplanted mice 3 months after transplantation (**e**) or *Mafb*[f/f]::LysM-Cre and control mice (8–10 weeks old) (**f**). The data were analyzed with the Mann–Whitney U-test, and the results from two independent experiments were pooled. **g** A western blot analysis was performed to measure the C1q protein levels in the serum of 8-week-old WT and *Mafb*[−/−] mice. Lower panel, the data were normalized to the protein band of Ponceau S and are presented as the mean ± s.e.m. *\*\*p < 0.01* compared with *Mafb*[f/f] (Student's *t*-test). The data are from one experiment that was representative of at least two independent experiments. **h** THP-1 cells were transfected with either control siRNA (si-control) or MAFB siRNA (si-MAFB). The cells were stimulated with dexamethasone and IFNγ for 24 h after PMA stimulation. The data were normalized to HPRT mRNA and are presented as the mean ± s.e.m. *n* = 4 for each group. *\*p < 0.05* compared with si-control (*Mafb, C1qa, C1qc*, Student's *t*-test; *C1qb*, Welch's *t*-test). The data are from one experiment that was representative of two independent experiments. **i** qRT-PCR was conducted using heterozygous *mafb*[b337/+] mutant zebrafish. The data were normalized to csf1r mRNA and are presented as the mean ± s.e.m. WT; *n* = 7, *mafb*[b337/+]; *n* = 6 for each group. *\*p < 0.05* compared with WT (Student's *t*-test). The data are from one experiment that was representative of two independent experiments

and immature dendritic cells (DC), and C1q in the serum is supplied to a sufficient degree by these cells, in contrast to other complement components that are expressed by hepatocytes in the liver[14, 15]. A C1q gene cluster (*C1qa, C1qb*, and *C1qc*) is present in ~20-kb genomic regions and has synchronized expression, but the molecular mechanism of this regulation is unknown[16, 17].

Nuclear receptor transcription factors are involved in the metabolic and immune activities of macrophages by regulating target genes[18]. Liver X receptor (LXR) activation induces the expression of *Mertk* and *Gas6* and promotes apoptotic cell

clearance to suppress inflammatory pathways[19]. A deficiency in RXRα, PPARγ, or PPARδ induces autoimmune kidney disease due to the inhibition of complement C1q expression or other apoptotic cell recognition factors[20, 21]. Our previous data showed that the LXR/RXR directly regulates the expression of *Mafb* in macrophages[9]. In the present study, we find that the efferocytosis ability is reduced in *Mafb*-deficient macrophages, as are the mRNA levels of *C1qa, C1qb*, and *C1qc* in *Mafb*-deficient peritoneal macrophages (PMs). Furthermore, activation of the classical complement pathway is reduced in *Mafb*-deficient mice, and

these mice are susceptible to autoimmune-inducing conditions, such as irradiation. By contrast, macrophage-specific *Mafb* conditional knock-out mice have a weak autoimmune phenotype under normal conditions. These results indicate that MafB is important for the regulation of C1q gene expression and promotes multiple C1q functions, such as inhibiting autoimmune disease and promoting the classical pathway.

## Results

**MafB deficiency leads to defective efferocytosis by macrophages.** Our previous microarray data revealed a nearly 50% reduction in *C1qa* and *C1qb* expression (Supplementary Fig. 1)[9]. Because several studies reporting a defect in C1q showed failed apoptotic cell clearance, we hypothesized that a lack of MafB may affect efferocytosis[11, 22]. First, we induced apoptosis in Jurkat cells or thymocytes with dexamethasone, and then the apoptotic stage was assessed using flow cytometry with 7AAD and an anti-Annexin V antibody. The early-apoptotic cells were positive for Annexin V and negative for 7AAD (Supplementary Fig. 2A). Next, to clarify the role of MafB in the efferocytosis by macrophages, we added early-apoptotic thymocytes to bone marrow-derived macrophages (BMDMs) or PMs from WT mice, and we found that *Mafb* expression was significantly increased in both types of macrophages (Fig. 1a). These data suggest that MafB induction is involved in the macrophage response to apoptotic cells. Next, we examined whether a lack of MafB affected the phagocytosis of apoptotic cells. We added fluorescent apoptotic Jurkat cells to WT or *Mafb*$^{-/-}$ fetal liver-derived macrophages and evaluated the ensuing phagocytosis by flow cytometry. The gating strategies are shown in Supplementary Fig. 3A–C. The results showed that the numbers of *Mafb*$^{-/-}$ macrophages that engulfed or bound the fluorescent apoptotic cells were significantly reduced at 30, 60, and 120 min compared with WT macrophages (Fig. 1b, c). We also examined whether *Mafb*$^{-/-}$ PMs and resident macrophages had the same phenotype as *Mafb*$^{-/-}$ fetal liver-derived macrophages. Because conventional *Mafb*-deficient mice die after birth, *Mafb*$^{-/-}$ and WT E14.5 fetal liver cells were transplanted into X-ray-irradiated recipient mice (8 weeks old) to generate hematopoietic-reconstituted mice. Apoptotic cells were injected into the abdominal cavity of the hematopoietic-reconstituted mice 2 months after transplantation. A fluorescence activated cell sorting (FACS) analysis performed 30 min after the injection of apoptotic thymocytes showed that the resident and thioglycolate-elicited PMs from *Mafb*$^{-/-}$ mice also failed to engulf or bind to fluorescent apoptotic cells (Supplementary Fig. 2C, D). To confirm that *Mafb*-deficient macrophages could neither bind to nor engulf apoptotic cells, we monitored phagocytosis using pHrodo-succinimidyl ester (pHrodo), which emits a fluorescent signal that becomes brighter in response to low pH (Supplementary Fig. 2B). After engulfment of the pHrodo-labeled apoptotic thymocytes, the fluorescent signal of pHrodo could be detected because of the acidic environment in the phagosomes of macrophages. This method prevents the detection of apoptotic cell binding on the surface of the macrophages[23]. Flow cytometry data showed a significantly reduced phagocytic efficiency of *Mafb*$^{-/-}$ fetal liver-derived macrophages (Fig. 1d, e) and PMs (Fig. 1g, h) compared with WT macrophages. Using fluorescence microscopy, we also observed that the intensity of pHrodo light emission from WT PMs was strong, whereas the signal intensity from the *Mafb*$^{-/-}$ macrophages was weak (Fig. 1f). These results demonstrate that MafB is indispensable for a large proportion of the phagocytosis of apoptotic cells by macrophages. By contrast, *Mafb*$^{-/-}$ macrophages could take up oxidized LDL and bacteria[9, 24], indicating that MafB deficiency specifically affects efferocytosis. In addition,

when we fed living thymocytes and fluorescent beads individually to WT or *Mafb*$^{-/-}$ macrophages, no difference was observed between the phagocytosis performed by WT or *Mafb*$^{-/-}$ macrophages (Supplementary Fig. 4A, B). This finding suggests that MafB regulates the efferocytosis process.

**MafB regulates C1q genes.** Because the *Mafb* deficiency led to a reduction in efferocytosis, we examined the expression of apoptotic cell recognition factors in *Mafb*$^{-/-}$ macrophages by qRT-PCR analysis. Consistent with the reduction in *C1q* our microarray results, the expression levels of *C1qa*, *C1qb*, and *C1qc* were significantly reduced in *Mafb*$^{-/-}$ fetal liver-derived macrophages (8-fold decrease in *C1qa*; 2-fold decreases in *C1qb* and *C1qc*, Supplementary Fig. 5A). A qRT-PCR analysis was performed using peritoneal exudative macrophages from *Mafb*$^{-/-}$ fetal liver-transplanted mice (20 weeks old), *Mafb*$^{f/f}$::LysM-Cre mice (8 weeks old), and age-matched control mice. The results showed that the expression levels of *C1qa*, *C1qb*, and *C1qc* were decreased 7–10-fold in *Mafb*$^{-/-}$ PMs compared with the WT control (Fig. 2a). We also generated *Mafb* conditional knock-out mice (*Mafb*$^{f/f}$) and crossed them with *Mafb*$^{f/f}$ and LysM-Cre knock-in mice (*Mafb*$^{f/f}$::LysM-Cre) specifically expressing Cre recombinase in myeloid cells (Supplementary Fig. 6A, B). The expression of *C1q* genes in the *Mafb*$^{f/f}$::LysM-Cre PMs was also strongly reduced (50-fold decrease in *C1qa*, 25-fold decrease in *C1qb*, and 100-fold decrease in *C1qc*), as was the expression of *Mafb* compared with the control (Fig. 2b). Western blot analysis showed that the C1q protein was also decreased in *Mafb*$^{-/-}$ and *Mafb*$^{f/f}$::LysM-Cre PMs compared with the control (Fig. 2c, full-length uncropped blots are shown in Supplementary Fig. 7A, B). Next, to examine the reduction in C1q gene expressions under physiological conditions, donor (Ly5.1 congenic)-derived macrophages (CD11b$^+$CD45.1$^+$) in *Mafb*$^{-/-}$-transplanted mice were sorted by FACS and analyzed by qRT-PCR. More than a 4–8-fold reduction in C1q gene expression was observed in the spleen macrophages of *Mafb*$^{-/-}$-transplanted mice (Supplementary Fig. 5B). We also evaluated the expression levels of *Mfge8*, *Mertk*, *Tyro3*, *Axl*, *Tim-4*, and *Itagv*, which remained unchanged in *Mafb*$^{-/-}$ fetal liver-derived macrophages (Supplementary Fig. 8A). Although the expression levels of *Gas6* and *Itgb3* were significantly decreased in fetal liver-derived macrophages, there were no significant changes in the expression of *Mfge8*, *Gas6*, *Itgαv* or *Itgb3* in *Mafb*$^{-/-}$ PMs (Supplementary Fig. 8B). In peritoneal resident macrophages, Tim-4 is a key protein in efferocytosis[25]. FACS data using an anti-Tim-4 antibody revealed no change in Tim-4 expression between WT and *Mafb*$^{-/-}$ macrophages (Supplementary Fig. 8C). These results indicate that MafB regulates the *C1q* genes in macrophages from fetal liver, spleen, and peritoneal tissue.

Because previous reports have shown that immature DCs are an additional source of C1q[15], we induced immature DCs and mDCs from E14.5 fetal liver cells with IL-4 and granulocyte macrophage colony-stimulating factor (GM-CSF) in the presence or absence of LPS and then performed qRT-PCR (Supplementary Fig. 9A). Although *Mafb* expression tended to decrease upon DC differentiation, C1q expression was diminished in both *Mafb*$^{-/-}$ immature DCs (443-fold in *C1qa*, 661-fold in *C1qb*, 358-fold in *C1qc*) and mDCs (36-fold in *C1qa*, 81-fold in *C1qb*, 53-fold in *C1qc*) and in macrophages compared with WT (Supplementary Fig. 9B, Fig. 2d). These results indicated that the *Mafb* deficiency resulted in a reduction in the C1q gene expression in immature DCs. The reduction in C1q in *Mafb*$^{-/-}$ was observed in macrophages, immature DCs, and mDCs, suggesting that MafB regulates the total C1q protein in the serum.

C1q is the first activation protein to initiate the classical complement pathway; therefore, we hypothesized that MafB is

important for the classical complement pathway. To test this hypothesis, we conducted a hemolysis assay. The results revealed a 41% reduction in hemolysis in $Mafb^{-/-}$ serum compared with WT (each mouse was 20 weeks old, Fig. 2e). Similarly, hemolysis of $Mafb^{f/f}$::LysM-Cre was decreased by 57% compared with $Mafb^{f/f}$ (each mouse was 8–10 weeks old, Fig. 2f). Moreover, we

examined serum C1q protein levels by western blot analysis. The results showed a 78% reduction in the protein levels of C1q in $Mafb^{f/f}$::LysM-Cre serum compared with the control (Fig. 2g, full-length uncropped blots are shown in Supplementary Fig. 7C). To summarize, MafB deficiency decreased the activity of the classical complement pathway. These findings demonstrated that MafB is

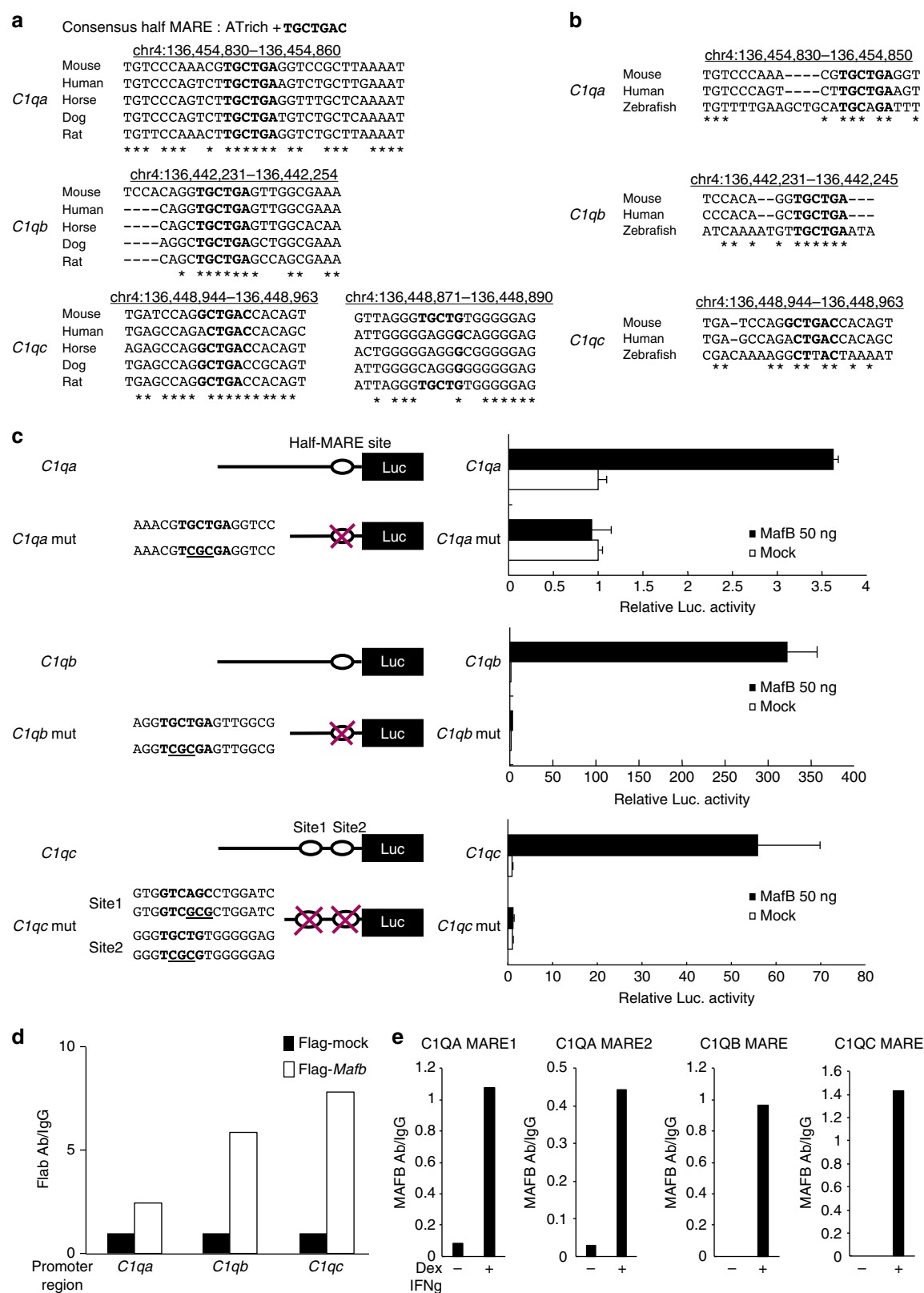

a key regulator of the classical complement pathway by regulating *C1q* genes in macrophages and DCs.

We also assessed the regulation of C1Q genes by MAFB in human macrophages. The human monocyte THP-1 cell line was cultured and induced to differentiated into macrophages by using phorbol myristate acetate (PMA). Although C1Q was not expressed in the THP-1 macrophages, C1Q genes were strongly induced by dexamethasone and IFNγ[26]. Under these conditions, *MAFB* expression was also induced in addition to C1Q genes (Supplementary Fig. 10). When *MAFB* expression was knocked down by siRNA, the expressions levels of *C1QA*, *C1QB*, and *C1QC* were significantly decreased (50% decreased in MAFB, 66% deceased in *C1QA*, 70% decreased in *C1QB*, 50% decreased in *C1QC*, Fig. 2h).

In addition, we examined whether MafB could regulate C1q expression in a lower vertebrate (zebrafish). Zebrafish also have three c1q genes (zc1qa, zc1qb, and zc1qc), which seem to function similarly to those in mammals[27]. To analyze the regulation of zebrafish c1q genes, we isolated RNA from adult tails of WT and *mafb*[b337/+], heterozygous MafB mutant fish with no DNA-binding domain[28], and we performed qRT-PCR. The results indicated that the expression of all three c1q genes was decreased in *mafb*[b337/+] mutants (50% decrease in zc1qa, 62% decrease in zc1qb, 63% decrease in zc1qc, Fig. 2i). Collectively, these findings suggest that the MafB-dependent regulation of C1q genes in macrophages is highly evolutionarily conserved among vertebrates.

Because the expression of *C1qa*, *C1qb*, *C1qc* was strongly decreased in *Mafb*-deficient macrophages in vitro and in vivo, and because our previous computational analysis study showed that the *C1qa* promoter contains a MARE motif and is activated by MafB overexpression[29], we hypothesized that MafB regulates all *C1q* genes by binding directly to the promoter. The sequences of the *C1qa*, *C1qb*, and *C1qc* promoters were analyzed to identify the putative transcription factor-binding sites for MafB. Notably, the potential half-MARE sites were identified at −131 bp upstream of the transcriptional initiation site of *C1qa*, −140 bp upstream of the transcriptional initiation site of *C1qb*, and +24 bp and +97 bp downstream of the transcriptional initiation site of *C1qc*. All the potential half-MARE sites were highly conserved among multiple mammalian species according to the UCSC Genome Browser (http://genome.ucsc.edu/index.html) (Fig. 3a), and they appeared to be conserved with respect to the C1q promoters in zebrafish (Fig. 3b). To evaluate the potential for MafB to bind to this half-MARE, we constructed luciferase reporter genes linked to the *C1qa*, *C1qb*, or *C1qc* promoters. These constructs were co-transfected with a MafB expressing vector into the RAW264.7 cell line. The results showed that the MafB-expressing vector activated the *C1qa*, *C1qb*, and *C1qc* promoters. Conversely, the *C1qa* −131/Luc mutant showed decreased responsiveness to the MafB-expressing vector (Fig. 3c, upper panel). On the *C1qb* promoter, the *C1qb* mut (−140)/Luc

reporter construct was not activated by MafB (Fig. 3c, middle panel). On the *C1qc* promoter, both single mutants (at *C1qc* +24/Luc and *C1qc* +97/Luc) showed decreased MafB-induced activation of the *C1qc* promoter. However, MafB could not activate the *C1qc* promoter with mutations (site 1: +97, site 2: +24, Fig. 3c, lower panel). To confirm direct MafB binding to these half-MAREs, we conducted a chromatin immunoprecipitation (ChIP) assay using the RAW 264.7 and human monocyte THP-1 cell lines. The results obtained for the RAW264.7 cells showed that MafB occupancy at these half-MAREs (*C1qa*: −131, *C1qb*: −140, and *C1qc*: +24 and +97) was increased after the transient transfection of the MafB expression vector in using anti-Flag antibody (Fig. 3d). Moreover, the anti-MAFB antibody detected endogenous MAFB occupancy, indicating direct binding to the conserved half-MAREs of the *C1QA*, *C1QB*, and *C1QC* promoters in dexamethasone-activated and IFNγ-activated human THP-1 cells (Fig. 3e). Another *C1QA* half-MARE (*C1QA* MARE 2, Fig. 3e) is not conserved in mice, and anti-MAFB also detected MAFB binding in this half-MARE. These results indicate that in both humans and mice, MafB directly regulates *C1qa*, *C1qb*, and *C1qc* through the half-MARE site.

Another Maf member, c-Maf, is also expressed in macrophages. Notably, the expression of *c-Maf* was maintained in *Mafb*[−/−] macrophages (Supplementary Fig. 11A). By contrast, the expression of *Mafb* and *C1qa* in *c-Maf*[−/−] macrophages was upregulated, suggesting that c-Maf could not induce C1q expression (Supplementary Fig. 11B). Consistently, the efferocytosis ability of *c-Maf*[−/−] macrophages seemed to be increased (Supplementary Fig. 11C). We also examined the expression of the C1q receptor (C1qRp) and another complement component known as C3 and its receptor, CR1, in *Mafb*-deficient macrophages. The expression of C1qRp was not reduced in *Mafb*[−/−] or *Mafb*[f/f]::LysM-Cre macrophages, while C3 and CR1 expression was slightly increased, suggesting a compensatory effect of the decrease in C1q (Supplementary Fig. 12A, B).

**C1q rescues efferocytosis defect of *Mafb*-deficient macrophages.** Based on the above results, we hypothesized that the efferocytosis defect in *Mafb*[−/−] macrophages may be caused by the reduction in C1q gene expression. To test this hypothesis, we rescued this defect using WT serum containing the C1q protein. The results showed that 50 and 20% dilutions of WT serum with medium rescued *Mafb*[−/−] efferocytosis in a dose-dependent manner (Fig. 4a). The uptake of apoptotic cells by *Mafb*[−/−] macrophages was improved in a time-dependent manner after the addition of 50% diluted WT serum for 60 and 120 min (Fig. 4b). Moreover, in heat-inactivated serum, the complement component function was disrupted, and it could not improve the defect in *Mafb*[−/−] efferocytosis (Fig. 4c). Furthermore, we used the serum

**Fig. 3** MafB directly regulates *C1qa*, *C1qb*, *C1qc* genes. **a** Half of the Maf recognition elements (half-MARE) were identified in C1q gene promoters using the UCSC Genome Browser. The half-MARE site in the C1q gene promoters (bold) was highly conserved among mammalian species. **b** Half-MARE was also conserved in zebrafish. **c** A MafB-expressing vector was co-transfected with luciferase reporter constructs of *C1qa*, *C1qb*, or *C1qc* promoter containing a MARE site or a mutant MARE site into RAW264.7 cells. The luciferase activity was analyzed after a 24 h transfection. The term "mut" indicates mutation in the MARE sites. Data are presented as the means ± s.e.m. of duplicates. The data are from one experiment that is representative of at least two independent experiments. **d** A ChIP assay was conducted using RAW264.7 cells. Flag-mock and Flag-MafB-expressing vectors were transfected into the cells. After 24 h, chromatin from the transfected cells was precipitated with anti-Flag and anti-IgG. The half-MARE sequences in the C1q gene promoters were amplified and analyzed by qPCR. The data are from one experiment that was representative of at two independent experiments. **e** A ChIP assay was conducted using THP-1 cells. The cells were collected after they were treated with or without dexamethasone (Dex) and IFNγ, and then chromatin was precipitated with anti-MAFB and anti-IgG. The half-MARE sequences in the C1q gene promoters were amplified and analyzed by qPCR. The data are from one experiment that was representative of two independent experiments

from *C1qa*-deficient mice. As expected, *C1qa*-deficient serum was also unable to rescue in the efferocytosis ability of *Mafb*^−/− macrophages (Fig. 4d). Finally, we examined whether purified C1q could rescue the *Mafb*-deficient macrophage phenotype. As expected, the percentage of *Mafb*-deficient macrophages with

apoptotic cells was significantly increased in the presence of C1q protein (Fig. 4e). The gating strategy of these data were shown in Supplementary Fig. 2D–G. Overall, these data indicate that the disruption of the efferocytosis ability of *Mafb*^−/− macrophages is caused by diminished C1q.

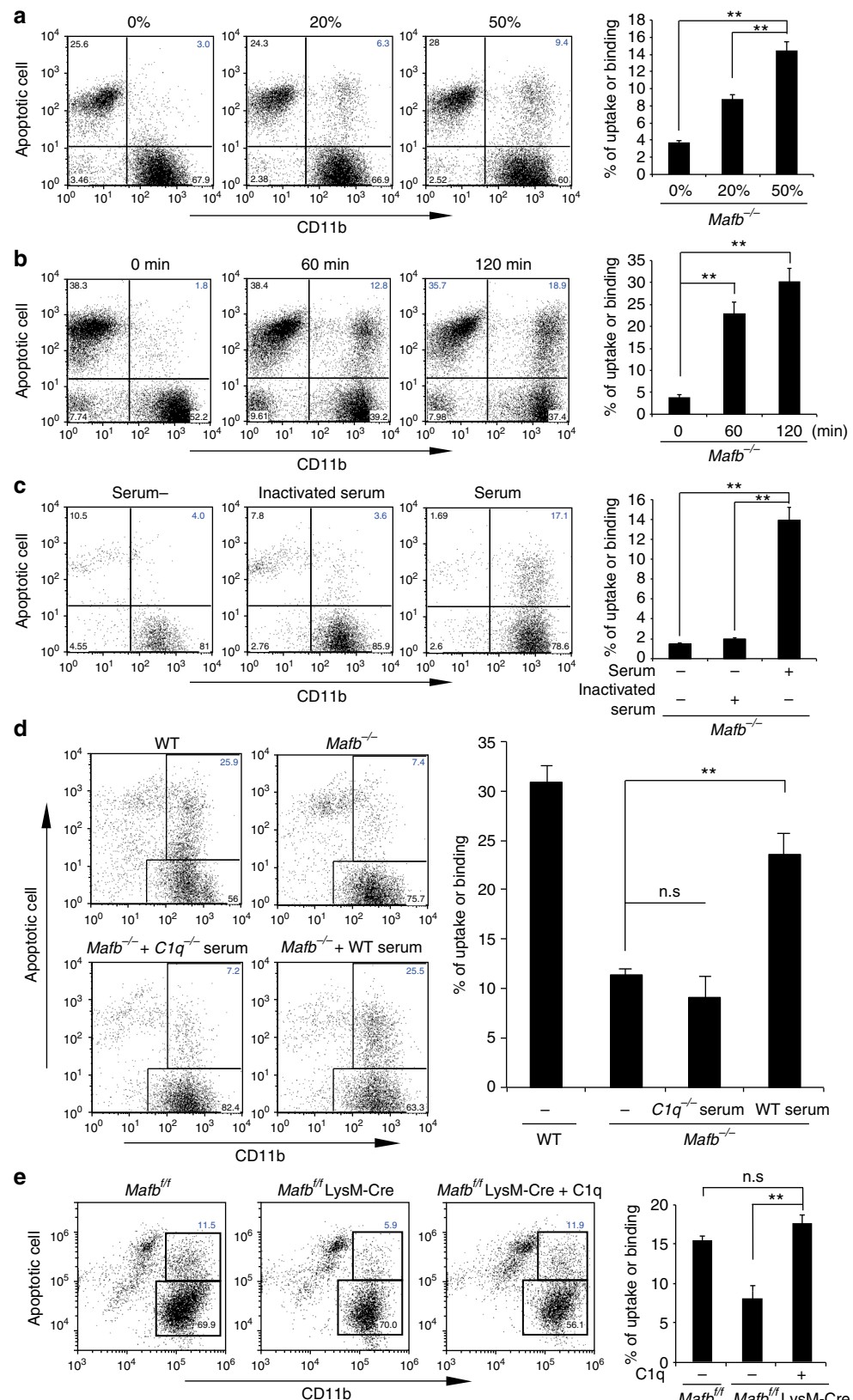

**Mafb-deficient mice develop autoimmune disease.** Because of the lack of C1q in both mice and human that were induced to have SLE-like autoimmune diseases[11, 12], we examined whether *Mafb*-deficient mice developed an autoimmune phenotype. In our fetal liver transplantation model, we used X-ray irradiation to destroy the hematopoietic system. Ionizing radiation is known to induce apoptosis[30–32], and apoptotic cells induce autoimmune disease[30, 33, 34]. Thus, we hypothesized that our transplanted recipient mice would have autoantibodies. We assessed the levels of anti-nuclear antibody (ANA) from 4 to 20 weeks after transplantation (mice aged 12 weeks to 28 weeks). Negative and positive signal of ANA titer is shown in Fig. 5a. Among the WT transplanted mice, 63% had high ANA titers at 4 weeks after transplantation. The ANA-positive samples then decreased in a time-dependent manner (8 weeks, 36%, 20 weeks, 30%; Fig. 5b). However, 52% of the *Mafb*$^{-/-}$ transplanted mice had a high ANA titer, and the percentage of mice with a significantly higher ANA titer remained at approximately 50% compared with WT (4 weeks, 47%, 20 weeks, 53%; Fig. 5b). At 20 weeks after transplantation, we also checked the titer range (1/160 to 1/2560). The results indicated that the ANA titer of the *Mafb*$^{-/-}$ transplanted mice had significantly increased ($p = 0.0395$, Fisher's exact test, Fig. 5b, c). ELISA data showed that more than half a year after fetal liver transplantation (30–32 weeks old), the serum levels of ANA and anti-dsDNA antibodies were significantly increased in *Mafb*$^{-/-}$ mice compared with the WT control mice (Fig. 6a, b). We also collected kidneys from these mice and performed periodic acid-Schiff (PAS) and Hematoxylin and Eosin (H&E) staining. More than 20 cases of glomerulonephritis from each genotype were observed and scored from 0 to 3 in a blinded test (Supplementary Fig. 13)[35]. The data showed that the average glomerulonephritis score of *Mafb*$^{-/-}$ mice (2.06) tended to be higher than that of the WT control (1.46), suggesting that the glomerulonephritis in *Mafb*$^{-/-}$ mice (34–38 weeks old) was slightly higher developed than that in WT control mice (Fig. 6e). Immunohistochemical staining of kidney sections showed that the accumulation of immune complexes (IgG, IgM, and IgA) on the glomeruli was higher in *Mafb*$^{-/-}$ mice than in WT mice (55–60 weeks old) (Fig. 6f, g). Moreover, urinary protein excretion in *Mafb*$^{-/-}$ mice was significantly increased compared with WT control mice (55–60 weeks old, $p < 0.05$, Welch's *t*-test, Fig. 6h). Conversely, the blood indices in *Mafb*$^{-/-}$ mice did not differ from those of WT mice, indicating that MafB deficiency did not affect hematopoietic cells (Supplementary Fig. 14). These results indicate that conditions that induce autoantibodies, such as X-ray irradiation, caused an accumulation of apoptotic cells and that *Mafb*-deficient mice may be more sensitive than the WT control. In addition, 4-month-old *Mafb*$^{f/f}$::LysM-Cre mice had slightly but significantly higher ANA and ds-DNA levels than did control mice (Fig. 6c, d). A previous study showed that C1q deficiency in a C57BL/6 genetic background did not show an ANA titer and have a severe autoimmune phenotype[36]. Our results are consistent with these findings because C57BL/6 background *Mafb*$^{f/f}$::LysM-Cre mice showed a weak phenotype. On the other hand, in autoimmunity-activated conditions, such

as the radiation-enhanced condition, autoimmune phenotype was observed in *Mafb*$^{-/-}$ mice. Thus, it is possible that one of the causes of the increased autoantibody levels in *Mafb*-deficient mice was a reduction in C1q in serum.

**MafB is a critical regulator of C1q-dependent gene expression.** Previous studies have shown that nuclear receptors, such as PPARδ, RXRα, RAR, and LXRα, regulate C1q-dependent gene expression and play roles in efferocytosis[20, 21, 37]. In particular, PPARδ binds directly to the *C1qb* promoter region and activates the expression of *C1qb*[20]. To compare C1q gene regulation between MafB and PPARδ, we performed a luciferase assay of the C1q promoters using a *Mafb*-expressing vector or a *PPARδ*/*RXRα*-expressing vector complex. The *Mafb*-expressing vector strongly induced the C1q promoters by approximately 40-fold (*C1qa*), 400-fold (*C1qb*), and 100-fold (*C1qc*). Conversely, the *PPARδ*/*RXRα*-expressing vector complex only induced a 2–3-fold increase in the C1q promoters (Fig. 7a). These data indicate that MafB can induce stronger activation of C1q promoters than PPARδ.

To compare their ability to induce endogenous C1q genes, we overexpressed *Ppard*/*Rxra*, *Pparg*/*Rxra*, *Lxra*/*Rxra*, *Rxra*, or *Mafb* in RAW264.7 cells. The data showed that *Mafb* overexpression strongly induced C1q-dependent gene expression, whereas PPARδ, PPARγ, LXRα, and RXRα were either not induced or were only slightly induced (Fig. 7b). Moreover, c-Maf expression did not induce C1q-dependent gene expression (Fig. 7b), confirming the different functions of c-Maf and MafB. Taken together, these data indicate that MafB plays a more prominent role than other nuclear transcription factors in the induction of C1q-dependent gene expression.

To determine whether these nuclear receptors (PPARδ, PPARγ, LXRα, and RAR) form a genetic hierarchy with MafB and C1q, we stimulated macrophages using agonists of PPARδ/RXRα, LXRα/RXRα, and RAR/RXRα, and we analyzed the expression of *Mafb*. Because previous studies have shown that glucocorticoids and IFNγ can induce the expression of C1q[16, 38], we also stimulated macrophages with dexamethasone or IFNγ. *Mafb* expression in macrophages was significantly induced by GW0742/9cRA, T1317/9cRA, ATRA, and dexamethasone. However, IFNγ did not induce the expression of *Mafb* (Fig. 7c). Next, we identified whether MafB was necessary for the induction of C1q expression by PPARδ/RXRα, LXRα/RXRα, RAR, dexamethasone, and IFNγ. *Mafb*-deficient and control macrophages were stimulated using GW0742 or T1317/9cRA for 16 h or with ATRA, dexamethasone, or IFNγ for 24 h. C1q expression was increased in both *Mafb*-deficient and control macrophages after stimulation with GW0742 (Fig. 7d). However, C1q-dependent gene expression was significantly lower in *Mafb*-deficient macrophages than in control macrophages. In contrast, T1317/9cRA, ATRA, and dexamethasone did not induce C1q expression in the *Mafb*-deficient macrophages, although *C1qc* expression was increased following ATRA stimulation (Fig. 7d). These results indicate that MafB is necessary for the regulation of C1q-

**Fig. 4** C1q reduction causes defective efferocytosis by *Mafb*$^{-/-}$ macrophages. **a** Apoptotic Jurkat cells treated with CellTracker were incubated with *Mafb*$^{-/-}$ fetal liver-derived macrophages in 0, 20, and 50% serum from WT mice (0%, *n* = 4; 20%, *n* = 3; 50%, *n* = 4). **b** Apoptotic Jurkat cells were incubated with *Mafb*$^{-/-}$ macrophages with 50% WT serum and a different time course (*n* = 4 for each group). **c** Apoptotic Jurkat cells were incubated with or without heat-inactivated serum or normal serum (serum-, *n* = 5; inactivated serum, *n* = 5; serum, *n* = 3). **a**–**c** The data are from one experiment that was representative of at least two independent experiments. **d** Apoptotic Jurkat cells were incubated with *Mafb*$^{-/-}$ macrophages with or without C1q-deficient serum or WT serum (WT, *n* = 6; *Mafb*$^{-/-}$, *n* = 10 for each sample). **e** Apoptotic Jurkat cells were incubated with *Mafb*$^{f/f}$::LysM-Cre fetal liver-derived macrophages that treated with or without 100 μg of purified human C1q 1 h before (*Mafb*$^{f/f}$, *n* = 3; *Mafb*$^{f/f}$::LysM-Cre, *n* = 10; *Mafb*$^{f/f}$::LysM-Cre + C1q, *n* = 10). **d**, **e** The results are from two pooled independent experiments. **a**–**e** Quantification data are presented as the mean ± s.e.m.; **p < 0.01, n.s. not significant (**a**–**e**, Welch's *t*-test)

dependent gene expression in the nuclear receptor pathway. In addition, PPARδ and IFNγ could induce low C1q-dependent gene expression in the absence of MafB, whereas MafB was necessary for C1q-dependent gene induction by LXRα/RXRα, RAR, and dexamethasone. Overall, MafB is necessary for C1q-dependent gene expression induced by nuclear receptor signaling.

## Discussion

We investigated, for the first time, whether MafB is an important factor in the expression of *C1qa*, *C1qb*, and *C1qc*, which assemble to form the C1q protein. C1q is the first protein in the classical complement pathway and is important for efferocytosis, which inhibits the development of autoimmune disease. More recent reports have shown that C1q binding to immune complexes also affects the inflammatory properties of CD14$^{dim}$ monocytes, and macrophages and DCs that have been polarized by C1q-late apoptotic lymphocytes reduce allogeneic Th1 and Th17 cells[39, 40]. These findings indicate that the prevention of autoimmune disease by C1q not only mediates efferocytosis but also regulates inflammatory suppression in macrophages and DCs. Because of a defect in these functions of C1q, *C1qa*-deficient mice develop autoimmune diseases[11, 22]. In addition, clinical reports from around the world have shown that 90% of patients with C1q mutations have autoimmune diseases, such as SLE[12]. Consistent with these reports, we observed a similar phenotype in *Mafb*-deficient mice. A previous report showed that the primary source of C1q is bone marrow-derived cells[41], and *C1q* genes are primarily expressed in macrophages and DCs[15, 41, 42]. Our results showed that *C1q* expression was decreased in both macrophages and DCs of *Mafb*-deficient mice. Moreover, the impaired classical pathway was observed because of decreased C1q protein in *Mafb*-deficient mouse serum. These results indicate that *Mafb* is important for C1q production to prevent autoimmune disease and enable innate immune responses.

According to data from microarray analyses of DC and macrophage subsets, *Mafb*, *C1qa*, *C1qb*, and *C1qc* have been identified as macrophage-specific genes[43]. These findings are consistent with our results, which indicate that MafB has a strong correlation with C1q expression. Therefore, we believe that MafB also regulates C1q genes in other species. In human macrophage THP-1 cells, knockdown of *MAFB* caused the downregulation of C1Q genes (Fig. 2h). Moreover, in zebrafish, c1q gene promoters have half-MAREs (Fig. 3b), and mutation of *mafb* revealed a reduction in c1q gene expression (Fig. 2i). Previous reports have demonstrated the origin of the classical pathway in the sea lamprey[44]. We also discovered a half-MARE sequence in the 5′ upstream region of lamprey c1q-like genes. Therefore, the MafB binding site may have contributed to the evolution of the more primitive lectin pathway to the classical pathway that supports adaptive immunity.

c-Maf, another large Maf transcription factor, is also expressed in macrophages. In this study, increased expression of *Mafb* and *C1qa* was observed in macrophage colony-stimulating factor (M-CSF)-induced *c-Maf*-deficient macrophages. Consistently, the phagocytosis of apoptotic cells by *c-Maf*$^{−/−}$ macrophages was increased compared with the WT control (Supplementary Fig. 11A–C). Therefore, the increased expression of the *C1qa* gene may be a secondary effect of *c-Maf* knock-out because of the increased expression of *Mafb*. These data, together with our previous results, demonstrate the different functions of MafB and c-Maf[3, 9, 45].

Several studies have investigated C1q gene promoters. In one paper, the *C1qb* promoter was activated by IFNγ, and the *C1qa* and *C1qc* promoters were suppressed. Results from a *C1qb*-deletion mutant promoter identified regulatory sequences from

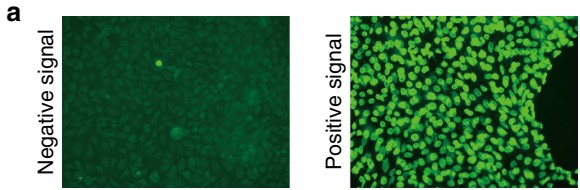

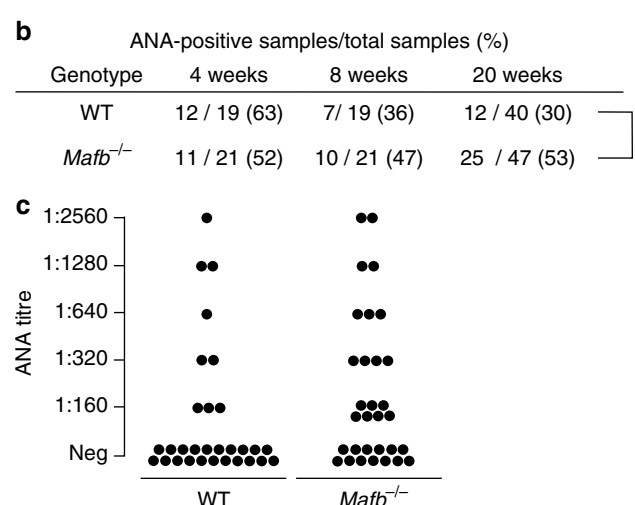

**b**

| | ANA-positive samples/total samples (%) | | |
| --- | --- | --- | --- |
| Genotype | 4 weeks | 8 weeks | 20 weeks |
| WT | 12 / 19 (63) | 7/ 19 (36) | 12 / 40 (30) |
| *Mafb*$^{−/−}$ | 11 / 21 (52) | 10 / 21 (47) | 25 / 47 (53) |

**Fig. 5** ANAs are detected in serum from fetal liver transplanted *Mafb*$^{−/−}$ mice. **a** Nuclei of HEp-2 cells were stained with mouse serum from WT or *Mafb*$^{−/−}$ fetal liver-transplanted mice using the HEPANA test. A positive signal indicates the presence of ANA. **b** Table showing ANA-positive numbers using serum diluted 1:60 and collected 4, 8, and 20 weeks after fetal liver transplantation; 4-8 weeks (male: WT, $n = 19$; *Mafb*$^{−/−}$, $n = 21$), 20 weeks (female: WT, $n = 12$; *Mafb*$^{−/−}$, $n = 15$; male: WT, $n = 28$; *Mafb*$^{−/−}$, $n = 32$). *$p < 0.05$ (Fisher's exact test). **c** Serum collected from WT or *Mafb*$^{−/−}$ mice 20 weeks after transplantation was diluted 160-fold, 320-fold, 640-fold, 1280-fold, and 2560-fold (WT, $n = 30$ (male, $n = 19$; female, $n = 11$); *Mafb*$^{−/−}$, $n = 31$ (male, $n = 17$; female, $n = 14$)). The number of *Mafb*$^{−/−}$ mice with an ANA titer (range: 1:160–1:2560) was significantly higher compared with WT mice ($p < 0.05$, Fisher's exact test). **b**, **c** Data from two to four independent experiments were pooled

−141 to −90[16]. These data appear to be consistent with our results showing a half-MARE in this region. Binding sites for IRF8 and PU.1 were observed in this promoter region, suggesting interactions between MafB and IRF8 or PU.1. However, further analyses are required.

Our previous study showed that MafB promotes atherosclerosis by regulating AIM expression[9]. However, C1q also has an important function in the inhibition of atherosclerosis. In early atherosclerosis, apoptosis by macrophages was increased in *Mafb*-deficient mice due to the reduction in AIM. Therefore, C1q may not effect efferocytosis in this condition. However, in other conditions, AIM also has multiple functions. It was recently reported that AIM can enhance the complement cascade[46]. Because both AIM and C1q bind to IgM, it is possible that AIM may enhance both the classical pathway and the alternative pathway during the development of hepatocellular carcinoma[46, 47]. AIM also inhibits obesity and ameliorates acute kidney injury, which suggests that it functions in homeostasis in the body[48, 49]. Because MafB regulates both AIM and C1q, MafB may be a regulator of homeostasis.

Under atherogenic conditions, the nuclear receptor transcription factor LXR regulates *Mafb*, which then regulates AIM[9]. In the present study, we showed that PPARδ/RXRα, LXRα/RXRα, RAR, or glucocorticoid receptor (GR) agonists could activate

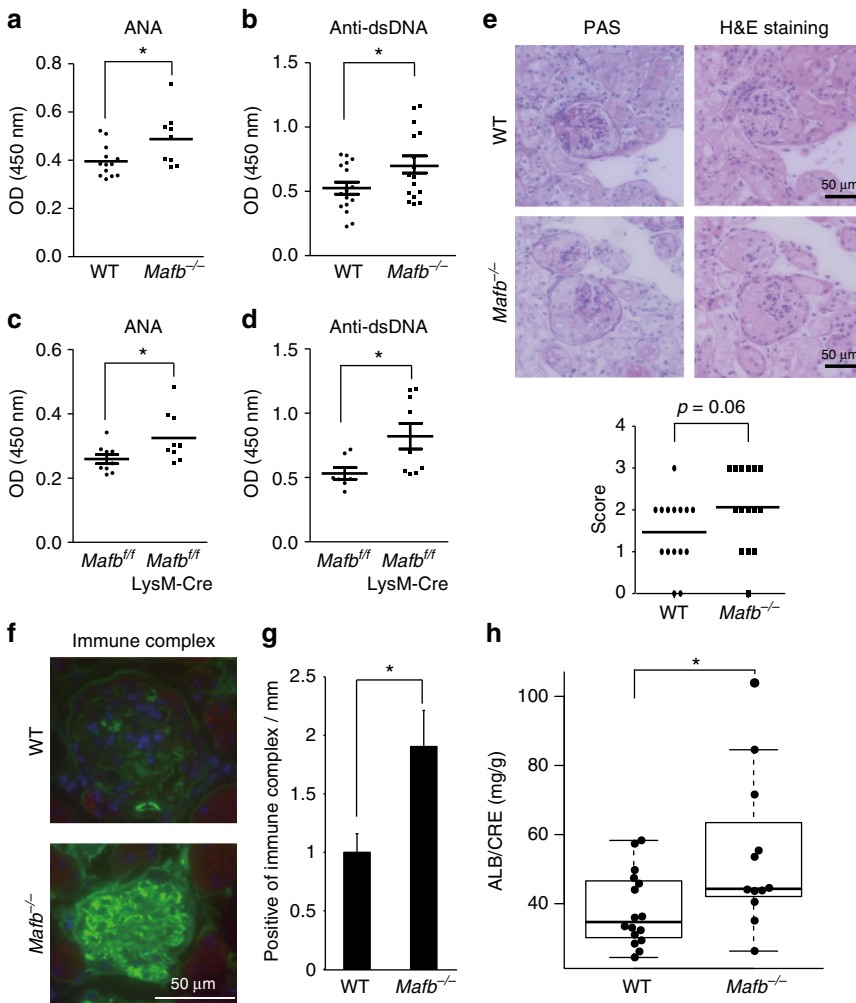

**Fig. 6** $Mafb^{-/-}$ mice develop autoimmune disease. **a** ELISA analysis showed the ANA levels in serum from 6 to 12-month-old transplanted mice (WT, $n = 13$; $Mafb^{-/-}$, $n = 9$). **b** The production of anti-ds-DNA was observed in serum ($n = 16$ for each genotype). **c** The ANA level of $Mafb^{f/f}$::LysM-Cre and $Mafb^{f/f}$ were analyzed by ELISA ($Mafb^{f/f}$, $n = 9$; $Mafb^{f/f}$::LysM-Cre, $n = 10$). **d** Serum anti-ds-DNA was analyzed by ELISA ($Mafb^{f/f}$, $n = 7$; $Mafb^{f/f}$::LysM-Cre, $n = 9$). **a–d** The results of duplicate independent experiments were pooled. **e** Kidneys were stained with PAS and HE staining. Glomerulonephritis was scored from 0 to 3 in a blinded test (WT, $n = 16$; $Mafb^{-/-}$, $n = 15$). The data were analyzed with the Brunner–Munzel test. **f** The deposition of immune complexes in glomeruli was assayed for the immune complex (IgG + IgM + IgA). **g** Immune complex-positive glomeruli were quantified (WT, $n = 5$; $Mafb^{-/-}$, $n = 5$). **h** Urinary protein was measured in WT and $Mafb^{-/-}$ 12-month-old transplanted mice (WT, $n = 17$; $Mafb^{-/-}$, $n = 12$). The data are presented as the mean ± s.e.m. *$p < 0.05$ (Welch's $t$-test). The results of two independent experiments were pooled

$Mafb$ expression. This evidence indicates that sensor molecules, which are members of the nuclear receptor family and include LXRs, RXRα, RARα, PPARδ, and GR, control macrophage-specific target genes by regulating $Mafb$. RXRs, which are partners of LXRs and PPARs, are considered druggable targets because of their functions in lipid metabolism and immunity. Although selective RXR modulators have been identified, few drugs for macrophage-associated diseases have been discovered because RXRs have multiple functions in various cells[50, 51]. Selective glucocorticoid receptor modulators have encountered similar problems in drug development[52]. The present study showed that the nuclear receptor-MafB-C1q pathway was macrophage-specific, which may facilitate the development of drugs for macrophage-related pathologies, such as autoimmune disease and atherosclerosis.

## Methods

**Mice**. The $Mafb^{-/-}$ mice were generated with a 129/Sv background and back-crossed to the C57BL/6J strain for seven or more generations[3]. The genotyping primer sequences were described in previous studies[3, 53]. For hematopoietic

system-reconstituted mice, $5 \times 10^6$ fetal liver cells were isolated from E14.5 WT or $Mafb^{-/-}$ (C57BL/6J-Ly5.1) embryos, and these cells were then injected into the tail veins of lethally irradiated (7 Gy) 6-week-old WT (C57BL/6J-Ly5.2) mice. The chimerism of the donor cells was determined based on Ly5.1$^+$/(Ly5.1$^+$ + Ly5.2$^+$) cell ratio. Mice with greater than 95% chimerism were used in further experiments. For $Mafb$ conditional knock-out mice, the $Mafb$ gene was flanked by a loxP element with a neomycin-resistant gene using homologous recombination in C57BL/6 background ES cells[54]. These mice were then mated with mice expressing flippase (Supplementary Fig. 4)[55]. To delete $Mafb$ specifically in the macrophage lineage, we mated $Mafb^{f/f}$ mice with LysM-Cre mice (Jackson Laboratory, Bar Harbor, Maine, USA) with its expression under the control of the endogenous Lys2 promoter[56]. The mice were maintained under specific pathogen-free conditions in a laboratory animal resource center at the University of Tsukuba. All experiments were performed in compliance with relevant Japanese and institutional laws and guidelines and were approved by the University of Tsukuba animal ethics committee (authorization number 17–152).

**Zebrafish**. AB strains were used as WT zebrafish. The $mafb$ knock-out line $mafb^{b337}$ was purchased from the Zebrafish International Resource Center (Eugene, OR, USA). The $mafb^{b337}$ line has a single C-to-T point mutation in allele 337 that changes a glutamine codon to a stop codon and truncates the protein upstream of the DNA-binding domain[28]. Because $mafb^{b337}$ zebrafish larvae die between 6 and 9 days after fertilization[57], we used a heterozygous mutant for the analysis. After anesthetizing the animals with tricaine methanesulfonate (Nacalai,

14805-82), we cut the tails of $mafb^{b337}$ adults for the isolation of genomic DNA and RNA for genotyping and qRT-PCR, respectively. For genotyping, genomic DNA was amplified at the mutation sites by PCR using the specific primers listed in Supplementary Table 1, and then it was digested with *PvuII*.

**Apoptosis induction**. To induce apoptotic thymocytes, we collected $1 \times 10^8$ thymocytes from 5 to 7-week-old WT mice and cultured them in RPMI-1640 (Sigma) medium containing 10% FBS, 1% penicillin/streptomycin, and dexamethasone (0.07 μM) for 12 h. To induce apoptotic Jurkat cells, $1 \times 10^7$ Jurkat cells were incubated in RPMI-1640 medium supplemented with 1% penicillin/streptomycin

and 50 μM etoposide (Sigma) for 16 h. Apoptotic cells were detected with a PE Annexin V Apoptosis Detection Kit (BD, 559763).

**In vitro apoptotic cell phagocytosis assay**. We labeled apoptotic Jurkat cells with 5 μM CellTracker (Invitrogen) for 30 min at 37 °C. Labeled apoptotic thymocytes and PMs were then mixed at a 1:1 apoptotic thymocyte:macrophage ratio and incubated for 30, 60, and 120 min. Phagocytosis of the labeled apoptotic thymocytes was then evaluated by flow cytometry. To rescue the phagocytosis ability of $Mafb^{-/-}$ macrophages, a mixture of apoptotic thymocytes and $Mafb^{-/-}$ PMs was plated in cultured medium containing inactivated or activated serum or various

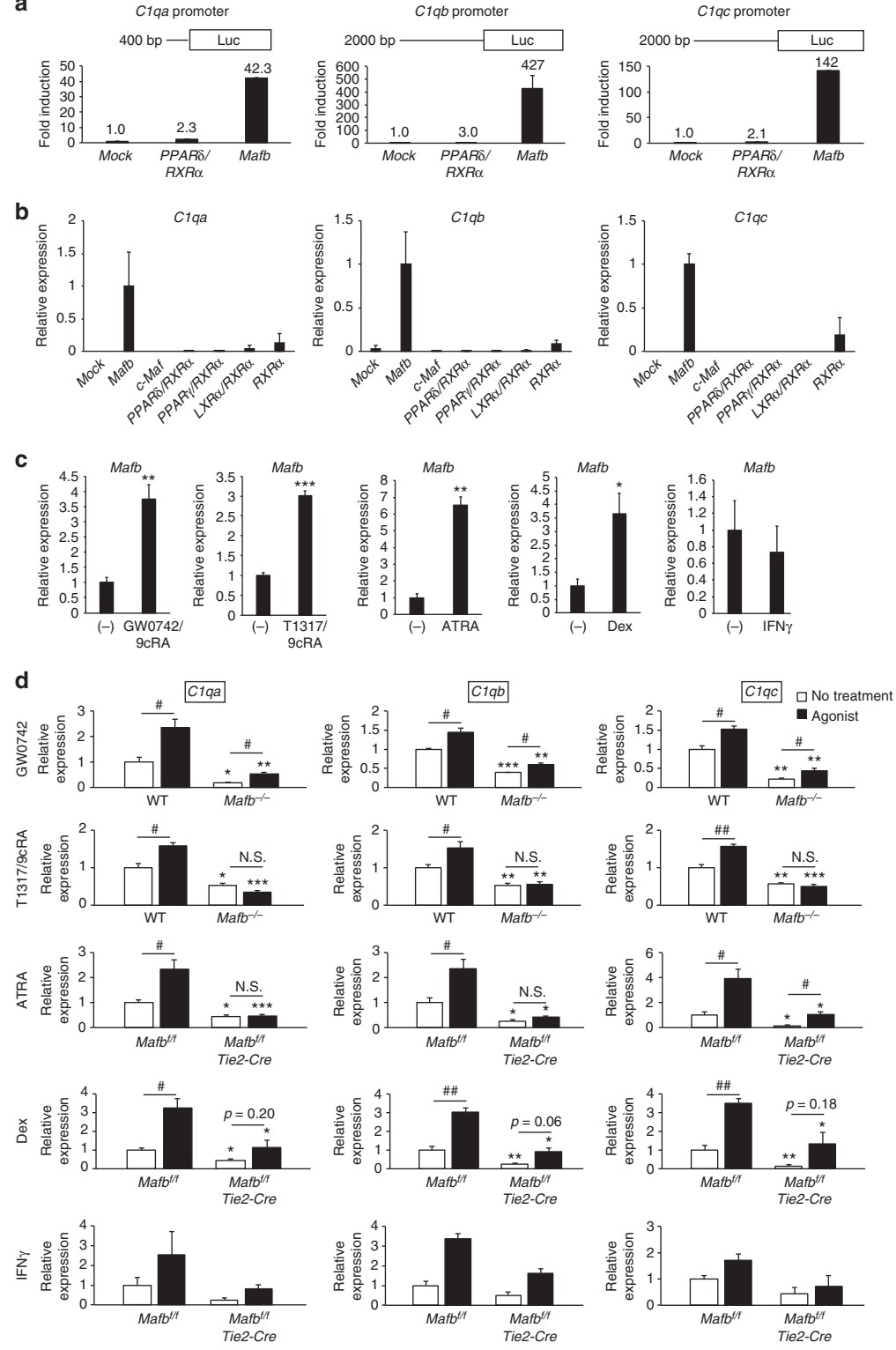

concentrations of serum (0, 20, or 50%) for 0, 60, or 120 min. In addition, serum from WT or $C1qa^{-/-}$ mice was used as a control for the rescue experiment. $C1qa^{-/-}$ serum was kindly provided by Professor Marina Botto (Centre for Complement and Inflammation Research, Imperial College, London). For the rescue experiment, 100 µg of purified human C1q (Complement Technology Inc.) was administered to fetal liver-derived macrophages with 400 µl of DMEM/F12 medium in the presence of M-CSF (10 ng ml$^{-1}$) 1 h before the addition of CellTracker-labeled apoptotic Jurkat cells. The cells were stained with antigen-presenting cell (APC)-conjugated anti-mouse CD11b antibody (Clone: M1/70, eBiosciences), and flow cytometry was performed with LSR (Becton Dickinson); the results were analyzed with FlowJo software (Treestar).

**In vitro fluorescent-bead phagocytosis assay**. The 1-µm fluorescent beads used in this experiment were Nile red fluorescent-conjugated carboxylate-modified microspheres (Molecular Probes, F-8819). Macrophages were mixed with 0.05% fluorescent beads for 2 h, and the cells were washed three times with PBS. Phagocytosis of the fluorescent beads was evaluated by flow cytometry with LSR (Becton Dickinson) and analyzed with FlowJo software (Treestar).

**In vivo apoptotic cell phagocytosis assay**. Apoptotic thymocytes were incubated with 20 ng ml$^{-1}$ pHrodo (Invitrogen) for 1 h at 37 °C. Then, the labeled apoptotic thymocytes were injected into the abdominal cavity of WT or $Mafb^{-/-}$ mice. After 30 min, PMs were collected and stained with APC-conjugated CD11b antibody, and phagocytosis was assessed by flow cytometry using LSR or a Biorevo fluorescence microscope (KEYENCE).

To assess both resident and elicited PM populations, a FACS analysis was performed using PE-conjugated anti-F4/80 antibody (Clone: Cl:A3-1, Adb Serotec), APC-conjugated anti-CD11b antibody (Clone: M1/70, eBioscience), or PE-conjugated anti-Tim-4 antibody (Biolegend).

**Macrophage and DC culture**. The mouse macrophage RAW264.7 cell line was cultured in Dulbecco's Modified Eagle's Medium (DMEM) supplemented with 10% FBS and 1% penicillin/streptomycin[9]. For the overexpression experiment, 500 ng of empty, MafB, c-Maf, PPARδ/RXRα, PPARγ/RXRα, LXRα/RXRα, and RXRα-expressing vectors was transfected into RAW264.7 cells using the FuGENE6 transfection reagent (Promega) for 24 h. For the primary macrophage culture, single-cell suspensions of fetal liver and bone marrow cells were prepared by mechanical disruption (grinding with a syringe insert against a 70-mm nylon cell strainer; BD Biosciences). Cells were re-suspended in DMEM supplemented with 10% FBS, 1% penicillin/streptomycin, and 10 ng ml$^{-1}$ M-CSF (R&D Systems) and then seeded onto tissue culture dishes. The culture medium was not changed for the duration of the experiment. M-CSF (final concentration, 10 ng ml$^{-1}$) was added every day from day 4 onwards[9]. To collect PMs, we injected thioglycolate into the abdominal cavities of the mice for 4 days. Then, PBS was injected into the abdominal cavities of the mice. The PBS was then collected and centrifuged at 1000 rpm for 4 min at 4 °C. The cell pellet was plated in cultured medium overnight to remove any neutrophils (non-adherent cells). Resident macrophages were collected without thioglycolate injection. Control (WT or $Mafb^{f/f}$) and $Mafb$-deficient ($Mafb^{-/-}$ and $Mafb^{f/f}$::LysM-Cre) macrophages were stimulated with PPARδ agonist (100 nM GW0742 and 1 µM 9cRA) for 16 h; LXRα agonist (1 µM T1317 and 1 µM 9cRA) for 12 h; or RAR agonist (1 µM ATRA and 1 µM 9cRA), glucocorticoid (10 µM dexamethasone), 10 ng µl$^{-1}$ IFNγ or vehicle for 24 h. For DC culture, fetal liver monocytes were cultured in RPMI supplemented with 10% FBS, 1% penicillin/streptomycin, 10 ng ml$^{-1}$ GM-CSF (R&D Systems, 415-ML) and 10 ng ml$^{-1}$ interleukin (IL) 4 (Pepro Tech, 214-14). GM-CSF and IL4 were added to the medium daily. For human macrophage cultures, human monocyte cell line THP-1 cells were maintained in RPMI-1640 medium (Sigma, R8758) supplemented with 10% FBS and 1% penicillin/streptomycin. THP-1 monocytes were treated with 16.2 nM phorbol 12-myristate 13-acetate (PMA) (Sigma, 79346) for 2 days to induce differentiation into macrophages. The PMA was then removed, and the THP-1 macrophages were stimulated with 10 µM dexamethasone and 10 ng ml$^{-1}$ IFNγ for 2 days.

**ChIP and quantitative PCR**. For mouse cells, a Flag-MafB-expressing vector was transfected into RAW264.7 cells for 24 h using FuGENE6. For human samples, THP-1 cells were differentiated into macrophages. Cells were fixed with 1% formaldehyde for 8 min at room temperature. Then, fresh medium containing 200 mM glycine was added and incubated for 8 min at room temperature. The cells were washed with PBS. Buffer NP-40 (10 mM Tris-HCl (pH 8.0), 10 mM NaCl, and 0.5% NP-40) was added, and the cells were collected. The cells were resuspended and centrifuged at $1400 \times g$ for 5 min at 4 °C with lysis buffer LB1 (50 mM HEPES (pH 7.5), 140 mM NaCl, 1 mM EDTA, 10% glycerol, 0.5% NP-40, 0.25% Triton X-100, and protease inhibitor), then lysis buffer LB2 (10 mM Tris-HCl (pH 8.0), 200 mM NaCl, 1 mM EDTA, 0.5 mM EGTA, and protease inhibitor), and finally lysis buffer LB3 (10 mM Tris-HCl (pH 8.0), 300 mM NaCl, 1 mM EDTA, 0.5 mM EGTA, 0.1% sodium deoxycholate, 0.5% N-lauryl sarcosine, and protease inhibitor). The samples were sonicated to generate approximately 500-bp chromatin fragments. Buffer LB3 containing 10% Triton X-100 was added to the samples, which were then centrifuged at 15,000 rpm for 10 min at 4 °C. After the "input" samples were separated, the supernatant was diluted in LB3 buffer and incubated overnight with Dynabeads containing protein G conjugated to anti-IgG (Merck Millipore), anti-Trimethyl-Histone H3 (Lys4) ChIP validated antibody (Merck Millipore, 17–614) and anti-Flag antibodies (Sigma, F3165) for mouse samples, and with anti-IgG (Merck Millipore), anti-Trimethyl-Histone H3 (Lys4) ChIP validated antibody (Merck Millipore, 17–614) and anti-MAFB antibodies (P-20, Santa Cruz) for human samples. Next, the immunoprecipitated samples were washed sequentially with low-salt buffer (20 mM Tris-HCl (pH 8.0), 150 mM NaCl, 2 mM EDTA, 0.1% SDS, and 1% Triton X-100), high-salt buffer (20 mM Tris-HCl (pH 8.0), 400 mM NaCl, 2 mM EDTA, 0.1% SDS, and 1% Triton X-100), RIPA buffer (50 mM HEPES (pH 7.6), 500 mM LiCl, 1 mM EDTA, 1% NP-40, and 0.7% sodium deoxycholate), and 50 mM NaCl buffer. The chromatin fragments were eluted from the Dynabeads with elution buffer (50 mM Tris-HCl (pH 8.0), 10 mM EDTA, and 1% SDS) at 65 °C overnight. The samples were incubated with RNaseA and Protease K for 2 h at 55 °C to degrade the RNA and protein. The DNA was then purified and used for quantitative PCR analysis. The "input" samples were used as standard qPCR samples. The primer sequences were specific to the MARE sites of the C1q promoters (Supplementary Table 1).

**Quantitative RT-PCR analysis**. Total RNA was collected using the Isogen kit (Nippon Gene, 311-02501). cDNA was synthesized with a QuantiTect Reverse Transcription Kit (Qiagen). The mRNA levels of mouse *Mafb*, *C1qa*, *C1qb*, *C1qc*, *PPARδ*, *Tim4*, *Gas6*, *Mfge8*, *Itgav*, *Itgb3*, and *MARCO*; human *MAFB*, *C1QA*, *C1QB*, and *C1QC*; and zebrafish *csf-1*, *mafb*, *c1qa*, *c1qb*, and *c1qc* were examined using SYBR green PCR master mix (Takara Bio). The mRNA levels were normalized to the mouse *Hprt* or *Gapdh* or human *HPRT* mRNA level. The specific primer sequences are listed in Supplementary Table 1.

**Western blot analysis**. PMs were lysed with RIPA buffer and then centrifuged at 15,000 rpm for 5 min at 4 °C. The supernatant was immunoblotted with anti-C1q antibody (Abcam, Clone: 9A7, ab71089). For the serum analysis, the serum was treated with the Qproteome Murine Albumin Depletion Kit (Qiagen). The samples were immunoblotted with anti-C1q antibody (Abcam, Clone: 9A7, ab71089).

**Hemolytic assay for C1q activity**. Sheep red blood cells (SRBCs, Nisseizai, 10100002) were sensitized with anti-SBRC antibody (Denka Seiken, 430076) at a 1/800 dilution to form Ab-sensitized SRBCs. Mouse serum was diluted 1/10 (for hematopoietic system-reconstituted mice) or 1/50 (for conditional knock-out mice) with GVB$^{2+}$ (ionic Veronal-buffered saline containing 0.15 M Ca$^{2+}$, 1 M Mg$^{2+}$, and 1% gelatin). C1q-depleted human serum (Merck Millipore) was diluted 1/500. Then, 100 µl of diluted mouse serum was incubated with 900 µl of C1q-depleted serum and 400 µl of EA for 1 h at 37 °C. The mixture was then centrifuged at 2000 rpm for 10 min at 4 °C, and the supernatant was measured spectrophotometrically at 541 nm.

**Fig. 7** MafB critically regulates C1q-dependent gene expression. **a** Activation of the *C1q* promoters by MafB or the PPARδ and RXRα complex was compared using a luciferase assay. The data are from one experiment that was representative of two independent experiments. **b** Empty vector (mock), MafB, c-Maf, PPARδ/RXRα, PPARγ/RXRα, LXRα/RXRα, or RXRα-expressing vectors were transfected into RAW264.7 cells. The expression of *Mafb*, *C1qa*, *C1qb*, and *C1qc* was analyzed using qRT-PCR ($n = 3$ for each group). The data are from one experiment that was representative of at least two independent experiments. **c** Control macrophages were treated with GW0742/9cRA (PPARδ/RXRα agonist), T1317/9cRA (LXRα/RXRα agonist), ATRA (RAR agonist), dexamethasone (glucocorticoid), and INFγ. *Mafb* expression was analyzed using qRT-PCR ($n = 3$ for the GW0742, T1317/9cRA, ATRA, and dexamethasone groups; $n = 2$ for IFNγ). **d** WT and $Mafb^{-/-}$ macrophages were stimulated with GW0742 or T1317/9cRA. $Mafb^{f/f}$ and $Mafb^{f/f}$::Tie2-Cre macrophages were stimulated with ATRA, dexamethasone or IFNγ. The expression of *C1qa*, *C1qb*, and *C1qc* was analyzed using qRT-PCR ($n = 3$ for the GW0742, T1317/9cRA, ATRA, and dexamethasone groups; $n = 2$ for IFNγ). **b**–**d** The expression of target genes was normalized to *Hprt* mRNA. **c**, **d** The data are presented as the mean ± s.e.m. *WT vs. $Mafb^{-/-}$, $Mafb^{f/f}$ vs. $Mafb^{f/f}$::Tie2-Cre; # no treatment vs. agonist; *, #$p < 0.05$, **, ##$p < 0.01$, n.s. not significant (Student's *t*-test)

**FACS analysis**. To examine the transplanted mice for chimerism, peripheral blood cells were isolated from WT and $Mafb^{-/-}$ mice. The cells were incubated with antibody for 30 min. Anti-CD45.1 and anti-CD45.2 antibodies conjugated to APCs or phycoerythrin (PE) (Anti-CD45.1, Clone: A20; Anti-CD45.2, Clone: 104, Bio-Legend) were used in this experiment. Flow cytometry analysis was performed using LSR (Becton Dickinson), and the results were analyzed with FlowJo software (Treestar).

**RNA interference**. siRNAs of MAFB were synthesized with Stealth Select RNAi (Invitrogen). The Stealth RNAi Negative Control (Invitrogen) was used as the control. The siRNAs were transfected with RNAi Max (Invitrogen). The siRNA and RNAi Max were mixed together at a 1:1 ratio. The mixture was then added to the differentiated THP-1 cells at $2.5 \times 10^5$ cells/well. BLOCK-iT Alexa Fluor Red Fluorescent Oligo (Invitrogen) was used to evaluate the transfection efficiency. The siRNA sequences are listed in Supplementary Table 1.

**Promoter analysis**. The promoter regions of the C1q genes (*C1qa, C1qb*, and *C1qc*) and the promoter region of the *Mafb* gene were analyzed with the UCSC Genome Browser (https://genome.ucsc.edu/index.html). The 5.0 kb *C1qa*, 2.0 kb *C1qb*, and −2000 bp to +200 bp *C1qc* promoters from genomic DNA of C57BL/6 mice were cloned into TOPO vectors and amplified individually. The C1q gene promoters-TOPO vectors were isolated and purified with a Mini Prep kit (Invitrogen). The C1q gene promoters were ligated into a promoter-less luciferase vector (pGL4.10, Promega) using *Kpn*I and *Hind*III to generate luciferase reporter gene constructs (*C1qa/C1qb/C1qc* promoter plasmids). *C1qa/C1qb/C1qc* mut promoter plasmids were constructed from *C1qa/C1qb/C1qc* promoter plasmids using the QuikChange Site-Directed Mutagenesis Kit (Stratagene) with the primers described in Supplementary Table 1.

To examine whether MafB could activate the C1q promoters, the C1q promoter vectors were co-transfected with the *Mafb*-expressing vector into RAW264.7 cells. To compare the activation of C1q gene promoters under the stimulation of MafB or PPARδ and RXRα, the C1q gene promoter vectors were co-transfected with *Mafb* or *PPARδ* and an *RXRα*-expressing vector into RAW264.7 cells. The transfection was performed using FuGENE6 transfection reagent (Promega). The cells were then cultured for 24 h. The luciferase assay was performed with the Dual-Luciferase Reporter Assay System (Promega). Briefly, after transfection for 24 h, the cells were lysed, and 20 μl of cell lysate was transferred to 100 μl of Luciferase Assay Reagent II (LARII). The samples were then measured using a luminometer. A 100-μl quantity of Stop&Glo reagent was added to the samples, and they were again measured. The pRL-TK (Promega)-expressing vector was co-transfected into RAW264.7 cells. A pRL-TK vector expressing *Renilla reniformis* luciferase was used to normalize the transfection efficiency.

**Autoimmunity in mice**. For the ANA titer analysis, we used the FLUORO HEPANA test (MBL). The anti-dsDNA antibody was evaluated using ELISA. An HRP-conjugated rabbit anti-mouse IgG + IgA + IgM (H + L) antibody (Invitrogen, 61–6420) was used as the secondary antibody, and 3,3′,5,5′-tetramethylbenzidine-hydrogen peroxide (TMP-$H_2O_2$, Thermo) was used as the substrate. Renal pathology was assessed based on the glomerulonephritis level and the accumulation of immune complexes in the glomeruli. Briefly, fixed kidneys were embedded in paraffin or optimum cutting temperature compound to perform histological or immunohistochemical staining, respectively. Paraffin sections were stained with PAS or H&E and then analyzed by a blinded researcher. More than 20 glomeruli from kidneys from each genotype were observed. The degree of glomerular lesion was semiquantitatively estimated on a 0–3 scale, and the most frequent score from the more than 20 glomeruli observed was adopted as the representative score of the glomerular lesion (Supplementary Fig. 13)[35]. For the immunohistochemical staining of glomeruli, 5-μm frozen sections were stained with rabbit anti-mouse IgG + IgA + IgM antibody (Invitrogen, 61–6420). Multiple images at 100× magnification were collected and combined to generate an entire kidney section using Biorevo image analyzer software (Keyence). The stained glomeruli were then counted. The average number of antibody-positive glomeruli from two kidneys/mouse was normalized to the kidney area. Urinary albumin and creatinine were analyzed using immunonephelometry and enzyme tests, respectively (Oriental Yeast Co., Ltd). GraphPad Prism 5 (GraphPad Software) was used to analyze the data.

**Analysis of blood indices**. Six months after transplantation, the blood from WT or $Mafb^{-/-}$ mice was collected in tubes containing EDTA. The blood was immediately analyzed using a Celltac-α automatic analyzer (NIHON KOHDEN, MEK-6458).

**Reagents**. The agonists used in this study were as follows: GW0472 (Sigma); T0901317 (T1317, Sigma); 9cRA (Sigma); dexamethasone (Sigma); ATRA (Sigma); and IFNγ (R&D Systems).

**Statistical analyses**. All data were analyzed using the Shapiro–Wilk test for normally distributed or non-normally distributed data. The data were also analyzed using the *F*-test, Ansari–Bradley test, or Mood test to evaluate homoscedasticity or heteroscedasticity. Parametric data analyzed with the Shapiro–Wilk test that were found to be homoscedastic or heteroscedastic were analyzed with Student's *t*-test or Welch's *t*-test, respectively. Non-parametric data that were homoscedastic or heteroscedastic were analyzed with the Mann–Whitney U-test or Brunner–Munzel test, respectively. All data were analyzed using R software (http://www.R-project.org/).

**Data availability**. The authors declare that the data supporting the findings of this study are available within the article and its Supplementary Information files.

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

## Acknowledgements

We thank Drs Takashi Moriguchi, Tomomasa Yokomizo, and Kazuko Shibuya for their helpful discussions and for providing reagents. We also thank Liliane Fossati-Jimack and Marina Botto for providing mouse C1qa-deficient serum. This work was supported by a Grant-in-Aid for Scientific Research (26221004, 25860205, 23118504, 21220009, 1612131, 16k16398, 17J01243); the Genome Network Project from MEXT of Japan; and by grants from the Uehara Memorial Foundation, Takeda Science Foundation, the Takamatsunomiya Cancer Foundation (15–24724; M. Hamada), and the World Premier International Research Center Initiative JSPS, Japan.

## Author contributions

M.T.N.T., M.H., Y.T., T.U., H.J., T.U. and R.S. performed the mouse experiments. M.T.N.T., M.H., R.S., K.A., M.H., Y.I., R.F., H.J. and M.N. performed the biochemical and cell biological experiments. M.T.N.T. and K.A. performed the ChIP analysis and hemolysis assay. M.T.N.T. and H.J. performed the western blot analysis. M.T.N.T., R.S., K.A., R.K., K.K. and Y.M. performed the qRT-PCR analysis. M.T.N.T., K.A., C.S.A. and M.K. performed the zebrafish experiment. R.K. and H.O. generated *Mafb* conditional knock-out mice. M.H., T.K., and S.T. contributed to the hypothesis development, experimental design, and data interpretation.

## Additional information

**Competing interests:** The authors declare no competing financial interests.

