## [Peer Review File · Nature Communications]

Reviewers' comments:

Reviewer #1 (Remarks to the Author):

The main thrust of the manuscript is that MafB directly controls the expression and production of C1q both in vitro as well as in vivo in mice. Consequently, the C1q-mediated effects on phagocytosis and recognition of apoptotic cells is affected.

The manuscript contains all the relevant data to support the view that MafB is essential in the production of C1q. Proper controls are included in each experiment.

The introduction of the manuscript deals with the relevant information on this issue, the methods section is well-detailed and the discussion takes care of the relevant points in relation to the role of MafB- regulated C1q antigen and function.

Minor point is: some more attention for a number of linguistic aspects. see for instance line 103 of the manuscript.

Reviewer #2 (Remarks to the Author):

This paper presents an important contribution to the understanding of the regulation of C1q synthesis, providing new information on this under investigated but potentially high impact area. Furthermore, using the findings of the regulation of C1q synthesis they provide further evidence for the importance of C1q functions in homeostasis and provide a potential molecular mechanism underlying the previous unexplained connection between the common activities induced by agonists of the PPAR and other members of this nuclear receptor family and activities mediated by C1q. The combined findings uniquely strengthen the understanding of the role of C1q in homeostasis, and suggests exciting new possibilities for regulation of auto-inflammation.

The major conclusions of this paper are that 1. MafB is a transcription factor that directly regulates the synthesis of C1q in both mice and human (with the demonstration of impairment in cells where the transcription factor binding sites area mutated). 2. MafB is downstream of, and necessary for, the nuclear receptor agonist (LXR α /RXR α , RAR and dexamethasone) upregulation of C1q synthesis, although there are other pathways which can also contribute to C1q synthesis. 3. MafB induction of C1q synthesis results in enhanced ability of macrophages for ingestion of apoptotic cells, and a deficient in MafB results in appearance of autoimmune pathologic features (immune complexes) and compromised renal function, supporting previous hypothesis of a role for C1q in suppression of autoimmune disorders.

While most of the studies are well designed and support the conclusions the set of experiments using the serum to rescue the defect in efferocytosis by MafB $^{-/-}$ macrophages is ambiguous. These experiments must be done using purified C1q (can be commercially obtained from Comptech) added alone (in the absence of other serum components) and added back into the C1q-deficient serum presented in Figure 3. This is important as in serum activation of complement results in many activation fragments (C5a, C3b, etc.) that could influence macrophage function/phagocytosis/efferocytosis and thus one cannot state that C1q is the lone mediator or not. In this regard, it is known from work of many others, more recently including Clarke, et al (1) and Elkon and colleagues (2), that ingestion of apoptotic cells may not be the critical activity, but rather the signaling by C1q in the macrophage itself leading to a suppression of the immune response to self antigens that may be the key C1q-mediated mechanism controlling autoimmunity. This should be mentioned in the Discussion of the paper.

The manuscript requires editing by an English speaking scientist, as there are several places where the text is either unclear or gives the wrong impression due to the choice of words or order of words or phrases. (Ex. Line 67-68 should be "binds antigen-bound antibody molecules leading to the activation of the classical complement pathway." Remove "types of ". Line 8 "is required for" must be replaced; line 140 and elsewhere "indispensable" should be changed or qualified by

adding "for a large proportion of the ...";182-3 – this sentence should be revised or moved to come after the data; 241 and elsewhere "large Maf" should be defined or large deleted;340- Ref 32 should be deleted as not relevant to the statement; 342-347 meaning unclear; 346-347 ; 365 "assumed"

From their own work, MafB^{-/-} macrophages are prone to apoptosis. How is can or does not impact the findings presented here should be discussed?

Age of the mice in the experiments should be noted. Quantitative degree of reduction of C1q expression should be written in the results section.

1 Clarke, E. V., Weist, B. M., Walsh, C. M. & Tenner, A. J. Complement protein C1q bound to apoptotic cells suppresses human macrophage and dendritic cell-mediated Th17 and Th1 T cell subset proliferation. *J.Leukoc.Biol.* 97, 147-160 (2015).

2 Santer, D. M., Wiedeman, A. E., Teal, T. H., Ghosh, P. & Elkon, K. B. Plasmacytoid dendritic cells and C1q differentially regulate inflammatory gene induction by lupus immune complexes. *Journal of Immunology* 188, 902-915 (2012).

Reviewer #3 (Remarks to the Author):

The authors analyzed the effects of MafB deficiency in macrophages on C1q expression, phagocytosis of dead cells (efferocytosis) and autoimmunity. They show that the level of MafB in macrophages is upregulated by the presence of apoptotic cells. Furthermore MafB deficient macrophages from different tissues, such as fetal liver derived, BMDMs or peritoneal (elicited or not by thioglycollate), were less capable to engulf apoptotic cells, but fetal liver derived MafB^{-/-} macrophages showed no difference in the uptake up fluorescent beads. Expression of C1q protein (WB) or transcripts levels of C1qa, C1qb and C1qc were also reduced in MafB^{-/-} macrophages from reconstituted irradiated mice or LysMcre/MafB^{f/f} mice. The authors also observed the same effect in zebrafish and THP-1 human cell lines. Using Chip-PCR they show that induced MafB bound to promoter regions of C1qa, C1qb and C1qc in macrophage cell lines. Using WT serum they were able to rescue the defective efferocytosis of MafB deficient macrophages in vitro, but he effect was abolished by heat inactivation and not present in C1q^{-/-} serum, suggesting a specific effect of C1q. Furthermore the authors report slight autoimmune symptoms in MafB^{-/-} fetal liver reconstituted chimeras and slightly increased serum autoantibodies in myeloid specific MafB deleted mice (using LysMCre).

Although In principle the question addressed in this manuscript is interesting, the paper remains rudimentary and key claims are not supported by robust data. Whereas the demonstration that MafB regulates the C1qa, C1qb and C1qc promoters is clear, it is not completely new, as this has been partially shown by the authors in an earlier manuscript that is not cited here. The claim that MafB primarily regulates C1q dependent gene expression or that it is the primary regulator of C1q is equally insufficiently supported by the data. Although it is demonstrated that the lack of C1q expression in MafB deficient macrophages affects their ability to phagocytose dead cells, the suggested link to autoimmune disease is very weak. The experiments of this section are limited and have problems in design, the effects are small and the statistical significance (as in some other parts) remains questionable. Taken together, the amount of clear and new data appears not sufficient for publication in a major general audience journal such as Nature Communications.

Major comments:

-Many observations lack robustness, in particular all experiments have been done only once except Figure 2H,I. Statistical analysis is missing in several parts (particularly supplementary figures) or is using inappropriate tests. Student's t-test used for most data sets is not appropriate for small populations and needs to be replaced by non-parametric methods such as Mann-Whitney test. In many cases insufficient experimental details are provided to judge the solidity of reported results

(some examples are indicated below).

- A potential autoimmune phenotype of MafB deficiency is not convincingly demonstrated. Although C1q deficiency has been proposed to increase susceptibility to auto-immune disease, the mechanisms are complex. To link reduced C1q expression in MafB deficient macrophages to an autoimmune phenotype in MafB deficient mice would therefore require careful analysis and robust effects. The data presented here, however, are unconvincing. All effects are small with poor statistical significance and many experiments are inappropriate in design.

o The increase of auto-antibodies shown in Fig.4A-D is marginal. No dilution curve has been done to properly dose the activity of auto-immune antibodies.

o Most importantly, it is unclear whether the (slightly higher than control) presence of autoantibodies is functionally important. This would require the demonstration that disease phenotype can be transmitted by serum transfer from MafB^{-/-} mice into WT mice.

o In addition the link of a claimed autoimmune phenotype of MafB deficient mice to reduced C1q expression is purely correlative. It would be required to demonstrate that any phenotype in MafB deficient mice could be rescued by injection of WT serum but not by C1q^{-/-} serum.

o The histology in Fig.4E is un-convincing. Differences appear marginal and the quantification shows a small effect and high p-value. The scoring criteria are not detailed. A cited reference refers to another study from the 60's. It is also not clear why WT mice already show a disease score, most of them of 2 or above. This suggests that MafB deficiency only slightly aggravates a pre-existing condition in the analyzed mouse colony rather than being causative.

o In this context it is also unclear exactly what WT controls were used. Are they coming from WT fetal livers of the same litter or a separate WT colony in the case of reconstituted mice? Were mice of the same age and sex reconstituted? This is important since a lupus-like disease can occur in 129 and C57BL/6 strains in the absence of any disrupted gene. For example a 129-derived interval on distal chromosome 1 has been strongly linked to autoantibody production (Heidiri et al. 2006). Although knockout mice were backcrossed, the exact status of controls and knockout mice in terms of presence of autoimmune disease susceptibility loci is not clear. Were LysMcre/+MafBf/f also backcrossed or only MafB^{-/-} fetal liver cells? The observation of variability and a high proportion of disease scores in WT mice suggest that autoimmune disease susceptibility loci might have been present in the scored control mice.

o It is unfortunate that not all autoimmune effects were also demonstrated in LysM-Cre MafBf/f mice to avoid potentially confounding or aggravating effects of BM reconstitution (irradiation, high concentration of cytokines and progenitors in the circulation, effects on the immune repertoire etc...). For LysM-Cre MafBf/f mice only small effects of increased autoantibodies are shown in fig.4C,D.

o It is unclear whether the quantification in fig .4F,G is significant. How many sections from how many mice were scored? How many glomeruli were counted? What level of IF intensity was considered positive?

o The difference in Fig.4H is not significant

-The concept that MafB activates the C1q promoter has already been published by the authors in 2011 in a study showing that induced MafB increased the proximal promoter of C1qa driven luciferase activity in RAW264.7 cells (Morita et al., BMC Systems Biology 2011) but the study is not cited in the current manuscript.

-The authors declare that total C1q protein in the serum is regulated by MafB based on the observation that C1q level is decreased in MafB^{-/-} macrophages and in induced DC. This analysis is incomplete. The authors only used iDCs, and one macrophage subset set in vivo, rather than sorted different classes of macrophage and DC subsets. They also didn't dose C1q serum protein levels (ELISA available).

It is misleading to claim that MafB primarily regulates C1q dependent gene expression. Work from multiple authors shows regulation of genes unrelated to C1q. Along similar lines it cannot be claimed that MafB is the primary regulator of C1q. Although the authors tested several different transcription factors the list is certainly not complete. Other Maf or AP-1 factors might be involved. The authors themselves showed a role for GATA-1 in negatively regulating the promoter in transient assays (Morita et al. 2011). In addition the authors only analyzed the proximal promoter elements. More distant enhancer elements might be involved that might be controlled by other

transcription factors. This would require genome wide analysis by ATAC or similar techniques to identify regulatory elements.

Specific comments:

-The materials and Methods are not complete.

-No explanation of the gating strategy used for all cells used or sorted: iDC, peritoneal macrophages (resident or elicited).

-Peritoneal macrophages (PM) are only sorted by CD11b+ staining but additional markers are necessary to clearly distinguish PM from other immunocytes and distinguish different PM sub-populations.

-Microarray data reported in the paper (line 103) were not shown in main or supplemental data (no evidence of the 50% reduction C1q in MafB deficiency in the paper that they mentioned).

-No explanation of LysM CRE origin? What is the ES origin used for the generation of the MafB floxed mice? (It's not detailed even in a previous article of the authors). Same comment for Tie2 Cre.

-No explanation of the Mutation in zMafB? Why is the level of Mafb the same in mafBb337/+? The primers used cannot discriminate the WT -zMafB and mut- zMafB?

-FigS5: Why didn't the authors test the same opsonin and receptors expression mRNA in A and B? Moreover, they claim that the level of those factors are not modulated by the absence of MafB. For instance, Tim4 is reduced in MafB deficient mice and could impact also efferocytosis.

-FigS10: In the title the authors claim that C1qRp is not reduced in MaB deficient macrophages but regarding the result in B, this is not convincing. In addition, no statistical test has been done.

Reviewers' comments:

Reviewer #1 (Remarks to the Author):

The main thrust of the manuscript is that MafB directly controls the expression and production of C1q both in vitro as well as in vivo in mice. Consequently, the C1q-mediated effects on phagocytosis and recognition of apoptotic cells is affected.

The manuscript contains all the relevant data to support the view that MafB is essential in the production of C1q. Proper controls are included in each experiment.

The introduction of the manuscript deals with the relevant information on this issue, the methods section is well-detailed and the discussion takes care of the relevant points in relation to the role of MafB- regulated C1q antigen and function.

Minor point is: some more attention for a number of linguistic aspects. see for instance line 103 of the manuscript.

Response: Thank you for your comment. Our manuscript has been revised by American Journal Experts, a professional manuscript-editing company. We have also added the results of a western blot analysis of C1q protein in serum (new Fig. 2) and ANA titers from a HEPANA test of WT and *Mafb*^{-/-} sera (new Supplementary Fig. 11). We believe that these data strengthen our hypothesis.

Reviewer #2 (Remarks to the Author):

This paper presents an important contribution to the understanding of the regulation of C1q synthesis, providing new information on this under investigated but potentially high impact area. Furthermore, using the findings of the regulation of C1q synthesis they provide further evidence for the importance of C1q functions in homeostasis and provide a potential molecular mechanism underlying the previous unexplained connection between the common activities induced by agonists of the PPAR and other members of this nuclear receptor family and activities mediated by C1q. The combined findings uniquely strengthen the understanding of the role of C1q in homeostasis, and suggests exciting new possibilities for regulation of auto-inflammation.

The major conclusions of this paper are that 1. Mafk is a transcription factor that directly regulates the synthesis of C1q in both mice and human (with the demonstration of impairment in cells where the transcription factor binding sites area mutated). 2. Mafk is downstream of, and necessary for, the nuclear receptor agonist (LXR α /RXR α , RAR and dexamethasone) upregulation of C1q synthesis, although there are other pathways which can also contribute to C1q synthesis. 3. Mafk induction of C1q synthesis results in enhanced ability of macrophages for ingestion of apoptotic cells, and a deficient in Mafk results in appearance of autoimmune pathologic features (immune complexes) and compromised renal function, supporting previous hypothesis of a role for C1q in suppression of autoimmune disorders.

While most of the studies are well designed and support the conclusions the set of experiments using the serum to rescue the defect in efferocytosis by *Mafb*^{-/-} macrophages is ambiguous. These experiments must be done using purified C1q (can be commercially obtained from Comptech) added alone (in the absence of other serum components) and added back into the C1q-deficient serum presented in Figure 3. This is important as in serum activation of complement results in many activation fragments (C5a, C3b, etc.) that could influence macrophage function/phagocytosis/efferocytosis and thus one cannot state that C1q is the lone mediator or not. In this regard, it is known from work of many others, more recently including Clarke, et al (1) and Elkon and colleagues (2), that ingestion of apoptotic cells may not be the critical activity, but rather the signaling by C1q in the macrophage itself leading to a suppression of the immune response to self antigens that may be the key C1q-mediated mechanism controlling autoimmunity. This should be mentioned in the Discussion of the paper.

Response: Thank you for your critical comments. We purchased purified C1q protein and used it to treat *Mafb*-deficient macrophages. As expected, the impaired efferocytosis was significantly rescued. We have included this result in the new Fig. 3E. We thank you for providing information regarding the recently described anti-inflammatory function of C1q. We agree with the reviewer's comment, and we understand that the inhibition of autoimmune disease by C1q occurs not only through the promotion of efferocytosis. We have added this information to the first paragraph of the Discussion section.

The manuscript requires editing by an English speaking scientist, as there are several places where the text is either unclear or gives the wrong impression due to the choice of words or order of words or phrases. (Ex. Line 67-68 should be "binds antigen-bound antibody molecules leading to the activation

of the classical complement pathway.” Remove “types of “. Line 8 “is required for” must be replaced; line 140 and elsewhere “indispensable” should be changed or qualified by adding “for a large proportion of the ...”; 182-3 – this sentence should be revised or moved to come after the data; 241 and elsewhere “large Maf” should be defined or large deleted; 340- Ref 32 should be deleted as not relevant to the statement; 342-347 meaning unclear; 346-347 ; 365 “assumed”

Response: Thank you for your helpful suggestions. We have modified these sentences according to the recommendations of the reviewer. Moreover, the manuscript has been edited by American Journal Experts (AJE), a company that specializes in English-language editing.

From their own work, MafB^{-/-} macrophages are prone to apoptosis. How is can or does not impact the findings presented here should be discussed?

Response: Thank you for your question. In our previous work, we showed that MafB regulates apoptosis inhibitor of macrophage (AIM) expression. Two papers have shown that AIM functions to inhibit apoptosis under specific conditions, such as the presence of atherosclerotic lesions or *Listeria monocytogenes* infection (Arai et al., 2005, *Cell Metabolism*, Joseph et al., *Cell*, 2004). Thus, we did not observe any changes in the number of MafB-deficient macrophages under normal conditions, indicating that macrophage apoptosis was not affected. Additionally, AIM has multiple functions, similar to C1q. It was recently reported that AIM can enhance the complement cascade (Maehara et al., *Cell Reports*, 2014). Because both AIM and C1q bind to IgM, it is possible AIM may enhance both the classical pathway and the alternative pathway during the development of hepatocellular carcinoma (Maehara et al., *Cell Reports*, 2014, Arai et al., *Cell Reports*, 2013). AIM also inhibits obesity and ameliorates acute kidney injury, which suggests that it functions in homeostasis in the body (Kurokawa et al., *Cell Metabolism*,

2010, Arai et al., *Nature Medicine*, 2016). Since MafB regulates both AIM and C1q, MafB may be a regulator of homeostasis.

We have added this information to the Discussion section.

Age of the mice in the experiments should be noted. Quantitative degree of reduction of C1q expression should be written in the results section.

Response: Thank you very much for your suggestion. We have added information about the age of the mice and the quantitative degree of reduction in C1q to the Results section. We think the inclusion of this information strengthens our paper.

1 Clarke, E. V., Weist, B. M., Walsh, C. M. & Tenner, A. J. Complement protein C1q bound to apoptotic cells suppresses human macrophage and dendritic cell-mediated Th17 and Th1 T cell subset proliferation. *J.Leukoc.Biol.* 97, 147-160 (2015).

2 Santer, D. M., Wiedeman, A. E., Teal, T. H., Ghosh, P. & Elkon, K. B. Plasmacytoid dendritic cells and C1q differentially regulate inflammatory gene induction by lupus immune complexes. *Journal of Immunology* 188, 902-915 (2012).

Reviewer #3 (Remarks to the Author):

The authors analyzed the effects of MafB deficiency in macrophages on C1q expression, phagocytosis of dead cells (efferocytosis) and autoimmunity. They show that the level of MafB in macrophages is upregulated by the presence of apoptotic cells. Furthermore MafB deficient macrophages from different tissues, such as fetal liver derived, BMDMs or peritoneal (elicited or not by thioglycollate), were less capable to engulf apoptotic cells, but fetal liver derived MafB^{-/-} macrophages showed no difference in the uptake up fluorescent beads. Expression of C1q protein (WB) or transcripts levels of C1qa, C1qb and C1qc were also reduced in MafB^{-/-} macrophages from reconstituted irradiated mice or LysMcre/MafB^{f/f} mice. The authors also observed the same effect in zebrafish and THP-1 human cell lines. Using Chip-PCR they show that induced MafB bound to promoter regions of C1qa, C1qb and C1qc in macrophage cell lines. Using WT serum they were able to rescue the defective efferocytosis of MafB deficient macrophages in vitro, but the effect was abolished by heat inactivation and not present in C1q^{-/-} serum, suggesting a specific effect of C1q. Furthermore the authors report slight autoimmune symptoms in MafB^{-/-} fetal liver reconstituted chimeras and slightly increased serum autoantibodies in myeloid specific MafB deleted mice (using LysMCre).

Although In principle the question addressed in this manuscript is interesting, the paper remains rudimentary and key claims are not supported by robust data. Whereas the demonstration that MafB regulates the C1qa, C1qb and C1qc promoters is clear, it is not completely new, as this has been partially shown by the authors in an earlier manuscript that is not cited here. The claim that MafB primarily regulates C1q dependent gene expression or that it is the primary regulator of C1q is equally insufficiently supported by the data. Although it is demonstrated that the lack of C1q expression in MafB

deficient macrophages affects their ability to phagocytose dead cells, the suggested link to autoimmune disease is very weak. The experiments of this section are limited and have problems in design, the effects are small and the statistical significance (as in some other parts) remains questionable. Taken together, the amount of clear and new data appears not sufficient for publication in a major general audience journal such as Nature Communications.

Major comments:

-Many observations lack robustness, in particular all experiments have been done only once except Figure 2H,I. Statistical analysis is missing in several parts (particularly supplementary figures) or is using inappropriate tests. Student's t-test used for most data sets is not appropriate for small populations and needs to be replaced by non-parametric methods such as Mann-Whitney test. In many cases insufficient experimental details are provided to judge the solidity of reported results (some examples are indicated below).

Response: Thank you for your critical comment. We very much appreciate this comment because we neglected to include the number of experiments in the figure legends. In the revised manuscript, we have added information regarding the number of experiments. Most of the presented results represent one experiment that is representative of at least two independent experiments or pooled results from more than 2 independent experiments. In addition, our experiments involved two mouse lines (conventional *Mafb* knockout mice and *Mafb* floxed mice::LysM Cre) or multiple analyses, such as qRT-PCR, western blotting, macrophage functional analysis, and hemolysis assays in serum. In addition, we analyzed not only mice but also human and zebrafish cells. Most of the data indicated that MafB regulates the *Clq* genes. Thus, we believe that our results are valid.

We also appreciate the reviewer's critique of our statistical analysis. In the revised manuscript, we have analyzed all the data using the Shapiro-Wilk test

to evaluate whether the data were normally or non-normally distributed. If the data were extremely non-normally distributed, we used a non-parametric test. The data were also analyzed using the F-test, Ansari-Bradley test, or Mood test to evaluate homoscedasticity or heteroscedasticity. Parametric data analyzed with the Shapiro-Wilk test that were found to be homoscedastic or heteroscedastic were analyzed with Student's t-test or Welch's t-test, respectively. Non-parametric data that were homoscedastic or heteroscedastic were analyzed with the Mann-Whitney U test or Brunner-Munzel test, respectively. All the data were analyzed using R software. We have added the name of the statistical test applied for each experiment in the figure legends.

- A potential autoimmune phenotype of MafB deficiency is not convincingly demonstrated. Although C1q deficiency has been proposed to increase susceptibility to auto-immune disease, the mechanisms are complex. To link reduced C1q expression in MafB deficient macrophages to an autoimmune phenotype in MafB deficient mice would therefore require careful analysis and robust effects. The data presented here, however, are unconvincing. All effects are small with poor statistical significance and many experiments are inappropriate in design.

Response: Thank you for your critical comment. We agree with the reviewer that the autoimmune phenotype of the *Mafb*-deficient mice was not strong. We believe this resulted from the genetic background because the C1q KO mice (in a C57BL/6 background) did not show a strong autoimmune phenotype. This is consistent with our results because our *Mafb*^{+/-} mice were backcrossed with C57BL/6 mice, and *Mafb*^{fl/fl} was generated using ES cells established from C57BL/6. In addition, irradiation caused the fetal liver-transplanted mice, even in a WT background, to develop autoimmune disease. Ionizing radiation is known induce numerous apoptotic cells (Potten and Grant, 1998; Sakaguchi et al., 1994; Takahashi et al., 2009). It is also known that apoptotic cells induce autoimmune disease (Henson and Bratton, 2013; Mevorach et al., 1998; Sakaguchi et al., 1994). Based on these findings,

we hypothesized that our transplanted recipient mice would produce autoantibodies. In the new figure S11A and B, we show the anti-nuclear antibody (ANA) titer results. Among WT transplanted mice 4 weeks after transplantation, 63% had an ANA titer (more than 1:160), which decreased in a time-dependent manner (30% in 20 weeks). However, the percentage of *Mafb*^{-/-}-transplanted mice with an ANA titer did not decrease. Additionally, a western blot analysis of serum using an anti-C1q antibody (new Fig. 2G) showed that C1q protein was strongly reduced in serum from *Mafb*-deficient mice. Consistently, a hemolysis assay using mouse serum showed that functional C1q was decreased in *Mafb*-deficient mice (Fig. 2E and F). These findings support the conclusion that C1q reduction was one cause of the development of autoimmunity in the *Mafb*-deficient mice.

o The increase of auto-antibodies shown in Fig.4A-D is marginal. No dilution curve has been done to properly dose the activity of auto-immune antibodies. Response: Thank you for your helpful comment. As we mentioned above, the transplanted mice developed autoantibodies due to induction by large numbers of apoptotic cells. We have added new ANA titer data to Figure S11C. The results showed a significantly greater number of *Mafb*^{-/-} mice with a high ANA titer (1:160 to 1:2560) compared with WT mice ($p < 0.05$, Fisher's exact test). Regarding Fig. 4C and D, we agree with the reviewer's comment. The difference in autoantibody levels was small. However, the development of autoimmunity in a C57BL/6 background is difficult, even in *C1q*-deficient mice; therefore, we do not think that these results are surprising.

o Most importantly, it is unclear whether the (slightly higher than control) presence of autoantibodies is functionally important. This would require the demonstration that disease phenotype can be transmitted by serum transfer from *Mafb*^{-/-} mice into WT mice.

Response: Thank you for your comment. We agree with the reviewer that this comment is very important. However, this experiment is quite difficult to perform because the WT recipients have abundant C1q protein in the serum. C1q is also continuously produced and functions to regulate immune suppression (Clarke et al., 2015; Santer et al., 2012). Therefore, the *Mafb* deficiency phenotype would be easily lost.

We believe our fetal liver transplantation model partially addresses this question because *Mafb*^{-/-} fetal liver cells were transplanted into WT recipient mice. Figure 2E shows that the function of C1q was reduced in the serum of these WT recipient mice. This result indicates that the *Mafb* deficiency phenotype was transmitted to the WT mice.

o In addition the link of a claimed autoimmune phenotype of MafB deficient mice to reduced C1q expression is purely correlative. It would be required to demonstrate that any phenotype in MafB deficient mice could be rescued by injection of WT serum but not by C1q^{-/-} serum.

Response: Thank you for your comment, which we believe is important. As mentioned above, the induction of autoimmune disease through the reduction of C1q is complex. Moreover, apoptotic inhibitor of macrophage (AIM) may play a role in this phenomenon. Because both AIM and C1q bind to IgM, the relationship between them should be considered. In fact, AIM controls the complement pathway (Maehara et al., 2014). Therefore, we believe the mechanism underlying autoantibody development in *Mafb* deficiency is also complex. In addition, C1q^{-/-} serum contains many activation fragments (C5a and C3b, among others) that could influence the function of macrophages. Therefore, after injecting WT or C1q^{-/-} serum, it is difficult to confirm that the C1q reduction was due solely to the production of autoantibodies in *Mafb*-deficient mice. The experiment itself is also technically difficult. A pre-experiment must be conducted to determine the required volume of serum

and the number of injections. Consequently, the experiment would take a long time.

Because a large body of research has already demonstrated that C1q reduction induces autoimmunity in both humans and mice, the purpose of this experiment is somewhat different from our purpose of determining whether MafB regulates the *C1q* genes.

o The histology in Fig.4E is un-convincing. Differences appear marginal and the quantification shows a small effect and high p-value. The scoring criteria are not detailed. A cited reference refers to another study from the 60's. It is also not clear why WT mice already show a disease score, most of them of 2 or above. This suggests that MafB deficiency only slightly aggravates a pre-existing condition in the analyzed mouse colony rather than being causative.

Response: Thank you for your comment. As mentioned above, irradiation may have caused the autoantibody production in this model (Supplementary Fig. 11B). Because the function of C1q is to inhibit or suppress the immune response (Clarke et al., 2015; Santer et al., 2012), it is not surprising that the phenotype was more aggravating than causative (Supplementary Fig. 11B)

o In this context it is also unclear exactly what WT controls were used. Are they coming from WT fetal livers of the same litter or a separate WT colony in the case of reconstituted mice? Were mice of the same age and sex reconstituted? This is important since a lupus-like disease can occur in 129 and C57BL/6 strains in the absence of any disrupted gene. For example a 129-derived interval on distal chromosome 1 has been strongly linked to autoantibody production (Heidiri et al. 2006). Although knockout mice were backcrossed, the exact status of controls and knockout mice in terms of presence of autoimmune disease susceptibility loci is not clear. Were *LysMcre/+MafBf/f* also backcrossed or only *MafB^{-/-}* fetal liver cells? The observation of variability and a high proportion of disease scores in WT mice

suggest that autoimmune disease susceptibility loci might have been present in the scored control mice.

Response: Thank you for your helpful comment. Regarding the fetal liver-transplanted mice, we used the same littermates of WT and *Mafb*^{-/-} mice, and every recipient mouse (8 weeks old) was purchased from the same company (Japan SLC, Inc.). As noted by the reviewer, it is also possible that the 129 locus persisted in our mouse model. Because C1q functions to inhibit autoimmune disease, this model is suitable for assessing the functions of C1q. Regarding the *Mafb*^{ff} mice, we used pure C57BL/6 ES cells (Tanimoto et al., 2008) to generate the *Mafb*^{ff} ES cells. LysM-Cre was also originally in the C57BL/6 background from the Jackson Laboratory, providing an explanation for the very weak phenotype of both the *Mafb*^{ff} LysM-Cre and C1q-deficient mice.

o It is unfortunate that not all autoimmune effects were also demonstrated in LysM-Cre *Mafb*^{f/f} mice to avoid potentially confounding or aggravating effects of BM reconstitution (irradiation, high concentration of cytokines and progenitors in the circulation, effects on the immune repertoire etc...). For LysM-Cre *Mafb*^{f/f} mice only small effects of increased autoantibodies are shown in fig.4C,D.

Response: Thank you for your comment. We think that C1q acts as an inhibitor after the induction of autoimmune disease. Regarding the reviewer's comment, our transplantation model may have been affected by irradiation-induced enhancement of autoimmunity. As shown in Supplementary Fig. 11B, both WT and *Mafb*^{-/-} mice had a high ANA titer exceeding 50% at 4 weeks after transplantation, but at 20 weeks after transplantation, the *Mafb*^{-/-} mice still had a high ANA titer. Considering the function of C1q to regulate immune suppression (Clarke et al., 2015; Santer et al., 2012), our findings are not unexpected. Based on the results from the *Mafb*^{-/-} and *Mafb*^{ff} LysM-Cre mice, the phenotype was stronger under

autoimmune-inductive conditions than normal conditions. The same phenomenon has been observed for C1q deficiency (Mitchell et al., 2002).

o It is unclear whether the quantification in fig .4F,G is significant. How many sections from how many mice were scored? How many glomeruli were counted? What level of IF intensity was considered positive?

Response: Thank you for your comment. We collected 2 kidneys from 1 mouse and fixed them in 4% PFA. Next, 1 section from 1 kidney was stained with a FITC-labeled anti-mouse IgG+IgA+IgM antibody. Multiple images at 10x magnification were collected and combined to generate an entire kidney section using Bioevo image analyzer software (Keyence). The stained glomeruli were then counted. Because it was difficult to determine whether the level of fluorescence intensity was positive, we distinguished between negative and positive according to the images shown in Fig. 4F. We counted two whole kidney sections/mouse. The average number of glomeruli from 2 kidneys was normalized to the area of the kidney. In WT mice, approximately 30 FITC-positive glomeruli per section were observed, while approximately 50 FITC-positive glomeruli per section were observed in *Mafb*^{-/-} mice. We have included this information in the Results section.

o The difference in Fig.4H is not significant

Response: Thank you for your comment. The data were analyzed using R software. The Shapiro-Wilk test and F-test showed that the data for both WT and *Mafb*^{-/-} were normally distributed and heteroscedastic. Therefore, they could be analyzed using Welch's t-test or the Brunner-Munzel test. The data were significant, as evidenced by a P value of 0.0406 (Welch's t-test) or 0.0402 (Brunner-Munzel test).

-The concept that MafB activates the C1q promoter has already been published by the authors in 2011 in a study showing that induced MafB increased the proximal promoter of C1qa driven luciferase activity in

RAW264.7 cells (Morita et al., BMC Systems Biology 2011) but the study is not cited in the current manuscript.

Response: Thank you for your comment. This paper is cited in our revised manuscript because the analysis supports our hypothesis. This analysis is also derived from the NCBI GEO datasets (GSE20419) deposited in February 2010. Morita et al. analyzed the ability of MafB over-expression to activate the C1qa promoter. This result triggered the initiation of the present study, in which we investigated MafB regulation of not only C1qa but also the C1qb and C1qc genes, directly *in vitro* and *in vivo* through a half-MARE sequence that is conserved from zebrafish to humans. In addition, this concept successfully led to the development of new concepts because the nuclear receptor transcription factors were confirmed to be upstream of MafB. Moreover, another target gene of MafB, AIM, was found to be important for homeostasis, especially under high-fat conditions. Both AIM and C1q have multi-homeostatic functions in macrophages, indicating that MafB may function as a regulator of homeostatic conditions during pathogenic events or in the presence of high cholesterol. Furthermore, we found half-MAREs in the promoter regions of zebrafish c1q genes and sea lamprey c1q-like genes, which suggested that the MafB binding site may have contributed to the evolution and establishment of the classical pathway supporting adaptive immunity. Therefore, we believe that our findings still have sufficient impact for publication.

-The authors declare that total C1q protein in the serum is regulated by MafB based on the observation that C1q level is decreased in MafB^{-/-} macrophages and in induced DC. This analysis is incomplete. The authors only used iDCs, and one macrophage subset set *in vivo*, rather than sorted different classes of macrophage and DC subsets. They also didn't dose C1q serum protein levels (ELISA available).

Response: Thank you for your critical comment. In the new Fig. 2G, we have added western blot results using serum from MafB^{ff} and MafB^{ff}::LysM-Cre

mice. The data showed a strong reduction in C1q expression in *Mafb^{ff}::LysM-Cre* serum. This antibody was strongly trapped by albumin protein; consequently, we depleted albumin prior to performing the western blot analysis, according to a previous study (Naito et al., 2012). An ELISA analysis was also performed with the same antibody used in the western blot analysis. However, the data were unstable, even for the WT samples, which we think is because the antibody detects serum albumin. The two types of western blot and hemolysis assay results indicated that C1q protein was significantly reduced in *Mafb*-deficient mouse serum. Regarding the DC analysis, Castellano et al. have shown that immature DCs express C1q, but fully mature DCs do not produce C1q. In addition, MafB expression was decreased in a mature DC subset (Satpathy et al., 2012) (Supplementary Fig. 6B). Moreover, cells expressing C1q in serum were not well identified. Therefore, it was difficult to sort the cells that contributed to C1q production in serum. Most importantly, the reduction in C1q was observed in serum, which indicated that the production of C1q was reduced in unknown C1q-expressing cells. Further analyses are required to identify these C1q-expressing cells.

It is misleading to claim that MafB primarily regulates C1q dependent gene expression. Work from multiple authors shows regulation of genes unrelated to C1q. Along similar lines it cannot be claimed that MafB is the primary regulator of C1q. Although the authors tested several different transcription factors the list is certainly not complete. Other Maf or AP-1 factors might be involved. The authors themselves showed a role for GATA-1 in negatively regulating the promoter in transient assays (Morita et al. 2011). In addition the authors only analyzed the proximal promoter elements. More distant enhancer elements might be involved that might be controlled by other transcription factors. This would require genome wide analysis by ATAC or similar techniques to identify regulatory elements.

Response: Thank you for your critical comment. We agree with the reviewer's opinion that the word 'primary' may be misleading. We do not wish to imply

that only MafB regulates *C1q* genes. Therefore, we have changed the word ‘primary’ to ‘main’.

Specific comments:

-The Materials and Methods are not complete.

-No explanation of the gating strategy used for all cells used or sorted: iDC, peritoneal macrophages (resident or elicited).

Response: Thank you for your comment. We have added the FACS profile for iDCs to Figure S6A and that for peritoneal macrophages (resident or elicited) to Supplementary Fig. 1C. We have also added the spleen macrophage gating strategy for sorting to Supplementary Fig. 3C.

-Peritoneal macrophages (PM) are only sorted by CD11b⁺ staining but additional markers are necessary to clearly distinguish PM from other immunocytes and distinguish different PM sub-populations.

Response: Thank you for your comment. Before the experiment, we confirmed that almost all the CD11b-positive macrophages were F4/80 positive. As shown in Supplementary Fig. 1C, the CD11b-positive cells were F4/80-positive in both elicited and resident macrophages.

-Microarray data reported in the paper (line 103) were not shown in main or supplemental data (no evidence of the 50% reduction *C1q* in MafB deficiency in the paper that they mentioned).

Response: Thank you for your comment. The microarray data are available in the NCBI GEO datasets (GSE20419). The analyzed data have been added to Supplementary Fig. 3A. *C1qa* and *C1qb* with a *Mafb*^{-/-}/WT microarray signal ratio less than 0.5 are presented.

-No explanation of LysM CRE origin? What is the ES origin used for the generation of the MafB floxed mice? (It's not detailed even in a previous article of the authors). Same comment for Tie2 Cre.

Response: The LysM-Cre mice were originally from the Jackson Laboratory. Cre is expressed under the control of the endogenous Lys2 promoter (Clausen et al., 1999). The ES origin of the MafB floxed mice was in a C57BL/6 background. We have added this information to the Methods section.

-No explanation of the Mutation in zMafB? Why is the level of Mafb the same in mafBb337/+? The primers used cannot discriminate the WT -zMafB and mut- zMafB?

Response: Information regarding zMafB is described in the Methods section. The *mafB*^{b337} line has a single C-to-T point mutation at position 337 in MafB that changes a glutamine codon to a stop codon and truncates the protein upstream of the DNA-binding domain. Because of the single point mutation, it is difficult to distinguish the mRNA of *mafB* and *mafB*^{b337}. Consequently, the levels of *mafB* were equivalent in *mafB*^{b337} and the control. For genotyping, because the mutation site contains a *PvuII* restriction enzyme site, the PCR product could be digested by *PvuII* to recognize mutant fish.

-FigS5: Why didn't the authors test the same opsonin and receptors expression mRNA in A and B? Moreover, they claim that the levels of those factors are not modulated by the absence of MafB. For instance, Tim4 is reduced in MafB deficient mice and could impact also efferocytosis.

Response: Thank you for your comment. We used fetal liver macrophages for the first screen (new Supplementary Fig. 5A). The Tim-4, ItgaV, Gas6, and Itgb3 genes were decreased, but the ItgaV, Gas6, and Itgb3 genes were not significantly reduced in the elicited macrophages. Tim4 is known to function in resident macrophages; therefore, we evaluated its expression by FACS. A reduction in Tim-4 was not observed in *MafB*^{-/-} resident macrophages (new Supplementary Fig. 5C). As shown in Fig. 2A, B, and C, only C1q genes were

strongly reduced (50-fold decrease in *Clqa*, 25-fold decrease in *Clqb*, and 100-fold decrease in *Clqc*). Therefore, we focused on the *Clq* genes. It is also possible that other opsonins also affected efferocytosis; however, this effect may be weaker than the effect of the C1q reduction.

-FigS10: In the title the authors claim that C1qRp is not reduced in MaB deficient macrophages but regarding the result in B, this is not convincing. In addition, no statistical test has been done.

Response: Thank you for your comment. We have changed the title. In addition, the C3 and CR1 levels were slightly elevated in the *Mafb*-deficient macrophages compared with the control macrophages, which may suggest a compensatory effect of the reduction in C1q. We have added this information to the Results section.

References (for Reviewer3)

Clarke, E.V., Weist, B.M., Walsh, C.M., and Tenner, A.J. (2015). Complement protein C1q bound to apoptotic cells suppresses human macrophage and dendritic cell-mediated Th17 and Th1 T cell subset proliferation. *J Leukoc Biol* 97, 147–160.

Clausen, B.E., Burkhardt, C., Reith, W., Renkawitz, R., and Förster, I. (1999). Conditional gene targeting in macrophages and granulocytes using *LysMcre* mice. *Transgenic Res.* 8, 265–277.

Henson, P.M., and Bratton, D.L. (2013). Antiinflammatory effects of apoptotic cells. *J Clin Invest* 123, 2773–2774.

Maehara, N., Arai, S., Mori, M., Iwamura, Y., Kurokawa, J., Kai, T., Kusunoki, S., Taniguchi, K., Ikeda, K., Ohara, O., et al. (2014). Circulating AIM prevents hepatocellular carcinoma through complement activation. *Cell Reports* 9, 61–74.

Mevorach, D., Zhou, J.L., Song, X., and Elkon, K.B. (1998). Systemic exposure to irradiated apoptotic cells induces autoantibody production. *J Exp Med* 188, 387–392.

Mitchell, D.A., Pickering, M.C., Warren, J., Fossati-Jimack, L., Cortes-Hernandez, J., Cook, H.T., Botto, M., and Walport, M.J. (2002). C1q deficiency and autoimmunity: the effects of genetic background on disease expression. *J Immunol* 168, 2538–2543.

Naito, A.T., Sumida, T., Nomura, S., Liu, M.L., and Higo, T. (2012). Complement C1q activates canonical Wnt signaling and promotes aging-related phenotypes. *Cell*.

Potten, C.S., and Grant, H.K. (1998). The relationship between ionizing radiation-induced apoptosis and stem cells in the small and large intestine. *Br J Cancer* 78, 993–1003.

Sakaguchi, N., Miyai, K., and Sakaguchi, S. (1994). Ionizing radiation and autoimmunity. Induction of autoimmune disease in mice by high dose fractionated total lymphoid irradiation and its prevention by inoculating normal T cells. *J Immunol* 152, 2586–2595.

Santer, D.M., Wiedeman, A.E., Teal, T.H., Ghosh, P., and Elkon, K.B. (2012). Plasmacytoid Dendritic Cells and C1q Differentially Regulate Inflammatory Gene Induction by Lupus Immune Complexes. *J Immunol* 188, 902–915.

Satpathy, A.T., Wu, X., Albring, J.C., and Murphy, K.M. (2012). Re (de) fining the dendritic cell lineage. *Nat Immunol*.

Takahashi, A., Ohnishi, K., Asakawa, I., Kondo, N., Nakagawa, H., Yonezawa, M., Tachibana, A., Matsumoto, H., and Ohnishi, T. (2009). Radiation response of apoptosis in C57BL/6N mouse spleen after whole-body irradiation. *International Journal of Radiation Biology* 77, 939–945.

Tanimoto, Y., Iijima, S., Hasegawa, Y., Suzuki, Y., Daitoku, Y., Mizuno, S., Ishige, T., Kudo, T., Takahashi, S., Kunita, S., et al. (2008). Embryonic stem cells derived from C57BL/6J and C57BL/6N mice. *Comp. Med.* 58, 347–352.

Reviewers' comments:

Reviewer #2 (Remarks to the Author):

The authors have presented generally well designed and performed experiments. This paper contributes to the understanding of the regulation of C1q gene expression and the role of C1q in homeostasis.

However, in response to the initial review, it is unclear why the previously suggested experiment of adding C1q back to show rescue of impaired efferocytosis was performed in the Mafb floxed mice (+/- LysM-Cre) where the uptake of apoptotic cells was very low in the cells with the floxed Mafb (relative to the wildtype shown in Figure 3D) and thus the window (1.9% in the control vs 1.3% positive cells in the Cre-deleted cells) is narrow. Wouldn't doing the experiment with Maf^{-/-} macrophages (as in Figure 3D) provided more convincing data? In addition, the amount of C1q added or how it was added (to the macrophages or to the apoptotic cells?) was not indicated in the M&M, results or Figure legends. The figure legend also did not indicate the n for the Mafb^{FL/FL} LysM-Cre + C1q condition. At a minimum, this information should be added to the manuscript.

Some edits are needed:

Line 190 – Should mDCs be added (to correspond to Figure)?

Line 201 – delete “to decrease”

Line 387 should be “bone marrow-derived cells”

Lines 390-393 should be revised or deleted. In contrast to what is stated there, decreased C1q protein in Mafb^{-/-} is not a consequence of impaired CCP but rather leads to the impairment.

Similarly, Mafb is important for C1q production to prevent autoimmune disease and enable innate immune responses.

Line 401 should be “reduction in c1q gene expression”

Line 967 should be “anti ds-DNA”

Source of LXR agonists, etc. should be added to M&M.generally

Reviewer #3 (Remarks to the Author):

The manuscript has significantly improved and useful new data have been added. Many details in methods and analysis have also been added and many questions have been correctly addressed in the rebuttal letter. The authors appear to agree with some of our previous criticism but then their comments and explanations did not result in the corresponding changes in the main manuscript. It is essential that the study's claims correspond to the shown data and the authors' own explanation in the rebuttal letter.

For example in the rebuttal letter the authors state that

- “ The (autoimmune) phenotype was more aggravating than causative”

- “ (the) very weak phenotype of both the Mafbf/f LysM-Cre”

- “ based on the results from the Mafb^{-/-} and Mafbf/f LysM-Cre mice, the phenotype was stronger under autoimmune-inductive conditions than normal conditions.”

This is not reflected in the introduction (line 95) or abstract, which states that “Mafb-deficient mice developed an autoimmune phenotype” and that “both Mafb-deficient fetal liver cells that were transplanted into recipient mice and macrophage-specific Mafb conditional knock-out mice exhibited autoimmune phenotypes”. This is misleading. For Mafbf/f LysM-Cre mice only very weak increases in autoantibody titers are shown and no glomerulitis or immune complex aggregation. The authors should clearly state that under normal conditions only these very weak phenotypes are observed and that transplantation of MafB deficient bone marrow extends the autoimmune effects of transplantation.

The authors also still state that "MafB PRIMARILY regulates C1q genes.." (line 50, 96). This may be a language issue, but as stated this ignores the role of MafB in the regulation of many other genes. The authors should modify this and say that "MafB is an important / critical regulator of C1q" or similar.

Similarly in the title it would be more correct to state that "MafB is A KEY/ CRITICAL regulator of C1Q" rather than "THE MAIN regulator". Other factors could be involved as well, especially via more distal enhancer elements that have not been investigated here.

The Materials and Methods are not complete. The protocol for MCSF induced BMDM differentiation should be added. The gating strategy for peritoneal FACS analysis should be added. In the WB section, please specify the Ab used for C1q.

The scoring criteria for glomerulonephritis used for Fig4E are still not well explained. The authors referred to Takahashi et al. 1991. For scoring this paper referred to an old paper that is not accessible via the internet (Suwa et al, 1964). Please clarify the exact scoring criteria you used on the Materials and Methods, especially since both the effects and differences are very small.

Minor comments:

- Modify figure legend G into H in Fig4. H is missing.

-In Suppl FigS1C: the PM cells profiles obtained on F4/80+ and Cd11b+ staining are quite unusual. There are no negative cells? This is even more surprising on elicited PMs, it means that you observed no monocytes infiltration?

-Indicate in the figure legend (Fig2C) that the western blot was done on peritoneal macrophage samples.

-Spelling mistake line 200

-Supplemental Fig3C: PMT for CD45.1 looks not appropriately set up.

Reviewer #2 (Remarks to the Author):

The authors have presented generally well designed and performed experiments. This paper contributes to the understanding of the regulation of C1q gene expression and the role of C1q in homeostasis.

However, in response to the initial review, it is unclear why the previously suggested experiment of adding C1q back to show rescue of impaired efferocytosis was performed in the *Mafb* floxed mice (+/- *LysM-Cre*) where the uptake of apoptotic cells was very low in the cells with the floxed *Mafb* (relative to the wildtype shown in Figure 3D) and thus the window (1.9% in the control vs 1.3% positive cells in the *Cre*-deleted cells) is narrow. Wouldn't doing the experiment with *Maf*^{-/-} macrophages (as in Figure 3D) provided more convincing data? In addition, the amount of C1q added or how it was added (to the macrophages or to the apoptotic cells?) was not indicated in the M&M, results or Figure legends. The figure legend also did not indicate the n for the *Mafb*^{FL/FL} *LysM-Cre* +C1q condition. At a minimum, this information should be added to the manuscript.

Thank you for your critical comment. With regard to the efferocytosis experiment, we attempted to use *Mafb*^{-/-} mice. However, because we typically use *Mafb* floxed mice, our *Mafb*^{+/-} mouse colony is small. Therefore, it was difficult to obtain sufficient fetal liver-derived macrophages in the allotted 3 months, as *Mafb* expression is strongly decreased in *Mafb*^{fl/fl} *LysM-Cre* (Fig. 2B). Theoretically, macrophages from *Mafb*^{-/-} and *Mafb*^{fl/fl} *LysM-Cre* mice should give the same result. Therefore, we chose to use *Mafb*^{fl/fl} *LysM-Cre* fetal liver-derived macrophages.

When we performed the experiments for the first revision, we had to establish the experimental system; however, the experiment conditions were considered inadequate. After changing the macrophage culture conditions, especially changing the FBS lot, the results were improved compared with the previous data. As shown in the new Figure 3E, fetal liver-derived macrophages from *Mafb*^{fl/fl} control uptake apoptotic cells (11.5%). Human C1q protein significantly rescued the phenotype of *Mafb*^{fl/fl} *LysM-Cre* macrophages (5.9% to 11.9%, n=10). Human C1q protein (100 µg) was added to the macrophages 1 hour before the addition of apoptotic cells in the presence of 10 ng/ml of M-CSF in DMEM/F12. We believe the new result is sufficient to support our hypothesis, and we have added this information to the Figure legends and Methods section.

Some edits are needed:

Line 190 – Should mDCs be added (to correspond to Figure)?

Thank you for your comment. We have added mDCs to this sentence.

Line 201 – delete “to decrease”

Thank you for your comment. We have deleted this text.

Line 387 should be “bone marrow-derived cells”

Thank you for your comment. We have changed the text to “bone marrow-derived cells”.

Lines 390-393 should be revised or deleted. In contrast to what is stated there, decreased C1q protein in *Mafb*^{-/-} is not a consequence of impaired CCP but rather leads to the impairment. Similarly, *Mafb* is important for C1q production to prevent autoimmune disease and enable innate immune responses.

Thank you for your comment. We have changed these sentences to “Moreover, the impaired classical pathway was observed because of decreased C1q protein in *Mafb*-deficient mouse serum. These results indicate that *Mafb* is important for C1q production to prevent autoimmune disease and enable innate immune responses.”.

Line 401 should be “reduction in c1q gene expression”

Thank you for your comment. We have changed the text to “reduction in c1q gene expression”.

Line 967 should be “anti ds-DNA”

Thank you for your comment. We have changed the text to “anti ds-DNA”.

Source of LXR agonists, etc. should be added to M&M.generally

Thank you for your comment. We have added the source of the agonists to the Methods section.

Reviewer #3 (Remarks to the Author):

The manuscript has significantly improved and useful new data have been added. Many details in methods and analysis have also been added and many questions have been correctly addressed in the rebuttal letter. The authors appear to agree with some of our previous criticism but then their comments and explanations did not result in the corresponding changes in the main manuscript. It is essential that the study's claims correspond to the shown data and the authors' own explanation in the rebuttal letter.

For example in the rebuttal letter the authors state that

- “ The (autoimmune) phenotype was more aggravating than causative”
- “ (the) very weak phenotype of both the *Mafbf/f LysM-Cre*”
- “ based on the results from the *Mafb-/-* and *Mafbf/f LysM-Cre* mice, the phenotype was stronger under autoimmune-inductive conditions than normal conditions.”

This is not reflected in the introduction (line 95) or abstract, which states that “*Mafb*-deficient mice developed an autoimmune phenotype” and that “both *Mafb*-deficient fetal liver cells that were transplanted into recipient mice and macrophage-specific *Mafb* conditional knock-out mice exhibited autoimmune phenotypes”. This is misleading. For *Mafbf/f LysM-Cre* mice only very weak increases in autoantibody titers are shown and no glomerulitis or immune complex aggregation. The authors should clearly state that under normal conditions only these very weak phenotypes are observed and that transplantation of *MafB* deficient bone marrow extends the autoimmune effects of transplantation.

Thank you for your critical comments, with which we completely agree.

In the abstract, we have changed the text from “During the adult stage, both *Mafb*-deficient fetal liver cells that were transplanted into recipient mice and macrophage-specific *Mafb* conditional knock-out mice exhibited autoimmune phenotypes.” to “During the adult stage, *Mafb*-deficient fetal liver cells that were transplanted into recipient mice exhibited autoimmune phenotypes in the autoimmune-inductive condition caused by X-ray irradiation.”

In the introduction, we have changed the text from “*Mafb*-deficient mice developed an autoimmune phenotype.” to “Moreover, *Mafb*-deficient mice could not suppress an autoimmune phenotype under an autoimmune-inducing condition, such as irradiation. In contrast, macrophage-specific *Mafb* conditional knock-out mice showed a weak autoimmune phenotype under normal conditions.”

The authors also still state that “*MafB* PRIMARILY regulates C1q genes.” (line 50, 96). This may be a language issue, but as stated this ignores the role of *MafB* in the regulation of many other genes. The authors should modify this and say that “*MafB* is an important / critical regulator of C1q” or similar.

Thank you for your critical comment. We understand that the term ‘primary’ has a strong meaning.

In the abstract, we have changed the text from “*MafB* primarily regulates C1q-dependent gene expression.” to “*MafB* is an important regulator of C1q.”

In the introduction, we have changed the text from “MafB primarily regulates C1q genes” to “MafB is important for the regulation of *C1q* gene”.

Similarly in the title it would be more correct to state that “MafB is A KEY/ CRITICAL regulator of C1Q” rather than “THE MAIN regulator”. Other factors could be involved as well, especially via more distal enhancer elements that have not be investigated here.

Thank you for your important comment. We agree with reviewer’s comments. We have changed the title to “MafB is critical regulator of complement component C1q”.

The Materials and Methods are not complete. The protocol for MCSF induced BMDM differentiation should be added. The gating strategy for peritoneal FACS analysis should be added. In the WB section, please specify the Ab used for C1q.

Thank you for your important comment. With regard to DMDM differentiation, we have added information regarding ‘Macrophage and dendritic (DC) cell culture’ to the Materials and Methods section.

With regard to the gating strategy for peritoneal FACS analysis, we have added Supplemental Fig. 1C with the gating method of the FSC and SSC panel.

With regard to the WB section, we used a C1q antibody (Abcam, ab71089) in both macrophage and serum data. We have added this information to the Methods section.

The scoring criteria for glomerulonephritis used for Fig4E are still not well explained. The authors referred to Takahashi et al. 1991. For scoring this paper referred to an old paper that is not accessible via the internet (Suwa et al, 1964). Please clarify the exact scoring criteria you used on the Materiel and Methods, especially since both the effects and differences are very small.

Thank you for your important comment. To explain the scoring criteria, we generated a new Supplementary Fig. 12. The image indicates capillary vessels in the glomerulus. Score 0 indicates normal architecture. Score 1 indicates increased mesangial substrate and infiltrative cells between capillary vessels. Score 2 indicates infiltrative cells into capillary vessels. Score 3 indicates disruption of the glomerulus structure. This information was added to the figure legend.

Minor comments:

- Modify figure legend G into H in Fig4. H is missing.

Thank you for your comment. We have corrected this error.

-In Suppl FigS1C: the PM cells profiles obtained on F4/80+ and Cd11b+ staining are quite unusual. There are no negative cells? This is even more surprising on elicited PMs, it means that you observed no monocytes infiltration?

Thank you for your comment. In these data, we changed the gating area to include the entire population (new Supplemental Fig. 1C). Then, negative cells were observed in both elicited and resident peritoneal macrophages. This finding indicates there may be monocyte infiltration.

-Indicate in the figure legend (Fig2C) that the western blot was done on peritoneal macrophage samples.

Thank you for your comment. We have added this information to the figure legend.

-Spelling mistake line 200

Thank you for your comment. We have changed 'smmarize' to 'summarize'.

-Supplemental Fig3C: PMT for CD45.1 looks not appropriately set up.

Thank you for your comment. We have added the gating strategy of this experiment in new Supplemental Fig. 3C.

In all of the FACS data in this paper, antibody activity was checked by comparing with the IgG isotype control or unstained control in every time experiment. Although the CD45.1 staining appears unclear, as the reviewer mentioned, the separation from CD45.2 host cells appears successful (Supplementary Fig. 3C gate c). The RT-PCR data were also consistent with other data that showed a reduction in C1q gene expression. Therefore, we believe that this experiment successfully supports our hypothesis.

REVIEWERS' COMMENTS:

Reviewer #2 (Remarks to the Author):

The authors have responded sufficiently to the comments of reviewers, and have presented generally well designed and performed experiments. This will be a important paper for the field, as new roles of C1q are discovered.

One further typo appears to be on line 403. 2H should be 2J?